



# A simple physical model for glacial cycles

Sergio Pérez-Montero[1,2], Jorge Alvarez-Solas[1,2], Jan Swierczek-Jereczek[2], Daniel Moreno-Parada[3], Marisa Montoya[1,2], and Alexander Robinson[4]

[1]Geosciences Institute (IGEO, CSIC-UCM), Madrid, Spain
[2]Complutense University of Madrid (UCM), Department of Earth Physics and Astrophysics, Madrid, Spain
[3]Université Libre de Bruxelles (ULB), Laboratoire de Glaciologie, Brussels, Belgium
[4]Alfred Wegener Institute, Helmholtz Centre for Polar and Marine Research, Potsdam, Germany

**Correspondence:** Sergio Pérez-Montero (sepere07@ucm.es) and Jorge Alvarez-Solas (alvarez.solas@igeo.ucm-csic.es)

**Abstract.** Glacial cycles are the norm in Pleistocene climate variability. Models of varying degree of complexity have been used to answer the question of what causes the nonlinear response of the climate system to the periodic forcing from the Sun. At one end of the spectrum of complexity are comprehensive models which aim to represent all involved processes in a realistic manner. However, their high computational cost precludes their use in the ultra-long simulations needed. At the other end are

conceptual models which are computationally less demanding but which generally lack a physical basis. Most of them yield good results in terms of capturing the shape and patterns of glacial cycles as indicated by the geological record, thus making it very difficult to identify the underlying mechanisms. Here we present a conceptual model that aims to physically represent the interaction between the climate and the Northern Hemisphere ice sheets while eliminating spatial dimensions in some of the fundamental ice-sheet thermodynamic and dynamic equations. To this end, we describe the Physical Adimensional Climate

Cryosphere mOdel (PACCO) from its simplest to its most complex configuration. We discuss separately the implications of different fundamental mechanisms such as ice-sheet dynamics and thermodynamics, glacial isostatic adjustment and ice-sheet albedo aging for our model. We conclude that ice-sheet dynamics and a delayed isostatic response are sufficient to produce resonance around periodicities of 100 kyr, although the forcing has a spectrum concentrated around lower values. In addition, ice-sheet thermodynamics and ice aging separately enhance the model nonlinearities to provide 100 kyr periodicities in good

agreement with reconstructions. However, we found that it is easier to reproduce the late Pleistocene glacial cycles using the simpler process of ice aging. Overall, PACCO is a valuable tool for analyzing the different hypotheses present in the literature.

## 1 Introduction

The climate variability of the Pleistocene, from 2.58 million years BP (before present) until today, is governed by the so-called Glacial-Interglacial Variability (GIV, Paillard, 2001, 2015; Ganopolski, 2024). Milankovitch (1941) postulated that this

variability results from changes in the insolation received by the Earth at the top of the atmosphere at 65ºN in boreal summer. Indeed, the main climate forcing at such long time scales results from the periodical variation of Earth's orbital parameters but the response of the climate system to such a forcing is not straightforward. Late Pleistocene GIV, from 800 to 11.7 kyr BP (thousand years), presents a marked 100-kyr periodicity and a "sawtooth pattern". This periodicity can be related to the





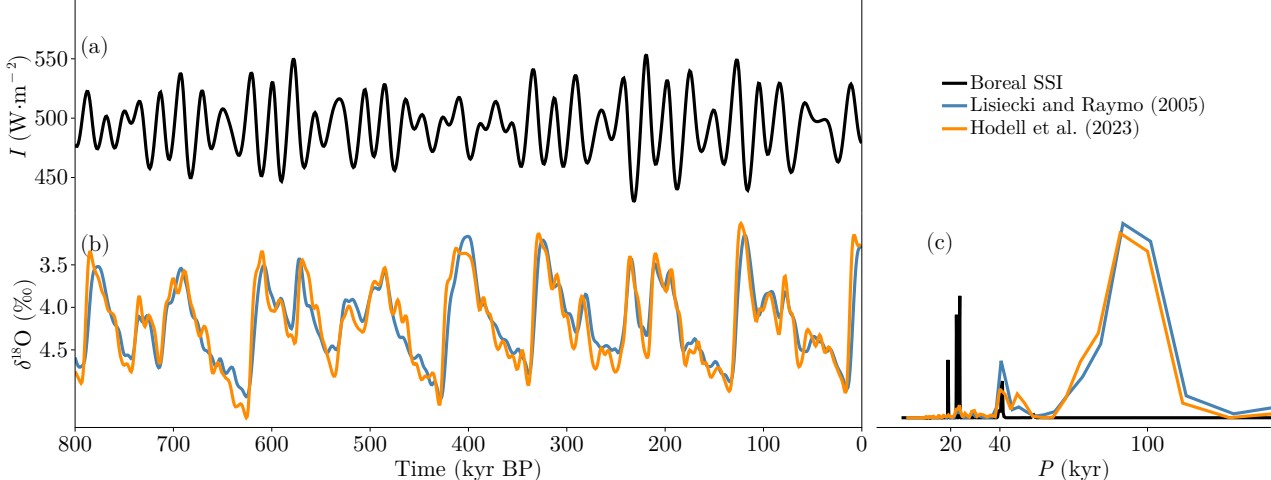

**Figure 1.** Time series of (a) boreal summer solstice insolation (SSI) at 65ºN following Berger (1978) and Laskar et al. (2004). (b) Oxygen isotope 18 ($\delta^{18}$O) from Lisiecki and Raymo (2005) and Hodell et al. (2023). (c) "100-kyr paradox" represented by the normalized periodogram for the last 800 kyr of time series shown in (a) and (b). All records were filtered with a lowpass Butterworth filter (cutoff frequency of 10 kyr$^{-1}$).

eccentricity of the orbit (∼100 and ∼400 kyr). However, the power of eccentricity in the insolation signal is negligible and the
effect is rather a modulation of the dominating precession cycle (∼20 kyr, Fig. 1, Berends et al., 2021a; Ganopolski, 2024),
consequently referred to as the "100-kyr paradox". Additionally the geological record reveals a sawtooth pattern (e.g., Lisiecki
and Raymo, 2005) that indicates a slow glaciation followed by a faster deglaciation or glacial termination. Both features suggest
a non-linear response of the Earth system to orbital changes.

Much effort has been devoted in the last decades to investigating the mechanisms responsible for the nonlinear nature of GIV
from a modeling perspective, both using conceptual and comprehensive models. The underlying hypotheses of these studies
are often of very different nature. Yet, most of them yield good results in terms of capturing the shape and patterns of glacial
cycles as indicated by the geological record, thus making it very difficult to identify precise mechanisms (Clark et al., 2006;
Imbrie et al., 2011; Paillard, 2015; Berends et al., 2021a; Verbitsky and Crucifix, 2023; Legrain et al., 2023; Ganopolski, 2024).

Within the conceptual modeling approach, much of the work done has involved mathematical models in the form of relax-
ation equations that reproduce GIV well. However, most of them rely on very mathematical approaches and include artificial
or imposed thresholds and trends (Paillard, 1998; Paillard and Parrenin, 2004; Gildor and Tziperman, 2001; Verbitsky et al.,
2018; Ganopolski, 2024). Paillard (1998) presented a three-state model based on insolation and ice volume thresholds. The
underlying hypothesis was that some part of the Earth system could provide the necessary nonlinear process in order to make
the climate system transition from one state to another. In particular, changes in ocean circulation were suggested to be the
driver of such nonlinearity, but these were not actually captured by the model. To solve this issue, Paillard and Parrenin (2004)
added an equation to the previous model that included the effect of oceanic stratification due to Antarctic sea ice and ice-sheet





extension. During a glacial phase, sea ice could grow far beyond the Antarctic coasts. Through brine rejection above continental margins deep water would become saltier and denser, favoring deep water formation and stratification of the Southern Ocean and thereby $CO_2$ storage. In this way, $CO_2$ levels would decrease in the atmosphere, allowing a colder climate and the

growth of the Antarctic ice sheet. When the Antarctic ice sheet reached its maximum extent at the limits of the continental shelves, sea-ice formation would be made further north and the deep water stratification would break down after a few thousand years, liberating high amounts of $CO_2$ that produce a glacial termination. This was suggested to be the main mechanism behind 100-kyr cycles.

Gildor and Tziperman (2001) instead proposed the so-called sea-ice-temperature-precipitation feedback as the main driver

of 100-yr cycles. In this case, the growth of the sea-ice extent at the end of glacial periods would inhibit precipitation over the North American ice sheet. In this way, the mass balance would decrease, allowing for glacial terminations. As a corollary, this hypothesis suggests that 100-kyr glacial cycles result from internal oscillations of the climate system rather than from forced response to an external forcing. However, as the authors stated, many physical processes were neglected so their results did not match the records. Later on, Verbitsky et al. (2018) built a model based on dimensional analysis of ice-sheet thermodynamics.

This model tried to represent the evolution of ice sheets via a linear relationship with climate temperature. When no forcing is applied, the model evolves to equilibrium. However, when forced, it reproduced different modes of rhythmicity depending on a dimensionless number (the variability number) defined as a function of eight parameters of the model. This number describes the relation between the negative and the positive feedbacks related to ice sheet basal sliding and temperature, respectively. If this criterion is large enough, the cycles are produced at ∼100 kyr due to multiples of obliquity and precession cycles. This

number also revealed that there is no need for non-linear relationships in the climate or in the carbon cycle in order to produce the rhythmicity observed in the paleo-records. However, the rather generalistic way by which it is defined, does not isolate the physical mechanism that produces the glacial-interglacial oscillations. Recently, an attempt was made to build a generalized Milankovitch Theory using conceptual models (Ganopolski, 2024). In this case, a model similar to that of Paillard (1998) was developed using results from the more comprehensive model CLIMBER-2 (Ganopolski and Calov, 2011). In this study, the

model satisfactorily reproduced GIV and a complete revision of the state of the art was made. Ganopolski (2024) highlighted nonlinearities associated with ice sheet size (increased basal velocities due to the presence of soft sediments, isostatic rebound, albedo darkening and enhanced melting due to proglacial lakes) as drivers of Pleistocene GIV.

Within the more comprehensive approach, Ganopolski and Calov (2011) employed CLIMBER-2 and investigated the role of $CO_2$ and the dust. The authors explained the late Pleistocene GIV as the consequence of the accumulation of dust on the surface

of ice sheets, thus increasing their sensitivity and lag with respect to insolation. They concluded that 100 kyr glacial stages are created when eccentricity is small enough to allow positive ice mass balance. When boreal summer insolation reaches high enough values (while increasing eccentricity), Northern Hemisphere ice sheets start decaying and low albedo enhances the response. In addition, they found that glacial terminations also require low $CO_2$ concentrations that amplify the cycles. Subsequently, Willeit et al. (2019) simulated the last 3 million years with CLIMBER-2 using multiple combined long-term

simulations. The authors found that glacial cycles are a quasi-deterministic response of their model to orbital forcing since the response is robust to initial conditions. This behavior is the result of regolith removal (the erosion of soft sediments beneath the





ice sheets), dust deposition and the gradual lowering of $CO_2$ as an imposed forcing trend. Therefore, dust was again identified as the trigger of the late Pleistocene GIV. Later, Mitsui et al. (2023) introduced a new mechanism called vibration-enhanced synchronization (after Pikovsky et al., 2003). There, the authors revisited the Quaternary glacial cycles of their model to

analyze this phenomena in detail. They found internal oscillations mainly related to dust and $CO_2$ feedbacks in agreement with the conclusions of Ganopolski and Calov (2011) and Willeit et al. (2019). Thus, if the internal frequency is similar to the forcing, a frequency entrainment (or synchronization) from the external forcing is possible. Then, if the internal oscillations (they found them to be around 95 kyr) synchronize with the climatic precession times when the eccentricity increases (since climatic precession is modulated by eccentricity) the system oscillates at ∼100 kyr. However, they found that their model could

be biased to glacial conditions since some deglaciations were not fully reproduced. On the other hand, Abe-Ouchi et al. (2013) used the comprehensive model IcIES-MIROC to study the role of isostatic rebound for glacial cycles. They found that the delayed response of the bedrock in the elevation-melting feedback on the ice sheets (after Oerlemans, 1980; Pollard, 1982) is a key process. They also found that ocean and dust feedbacks are not necessary and that glacial cycles are produced even with constant levels of $CO_2$, but that the amount of $CO_2$ amplifies the amplitude of the signal. Thus, the modulation of precessional

cycles by eccentricity was concluded to be the driver for the 100 kyr sawtooth glacial cycles.

To summarize, in the conceptual framework we can see two main problems when trying to solve Pleistocene physics: the lack of explicit physics and the need to impose ad-hoc thresholds. Most models do not explicitly solve the physical processes governing climate or ice sheets and need a change in their reference states via imposed thresholds to reproduce the GIV. In turn, the comprehensive approaches provide realistic and accurate results that shed light on the likely evolution of the Earth

system across the Pleistocene. However, the number of processes solved hinders the isolation of the mechanisms governing the GIV. Furthermore, despite the numerous hypotheses proposed and the wide range of model complexities employed, there is no definitive answer to the 100-kyr paradox. In this context, we have built a model with the minimum amount of explicitly-resolved physical processes that successfully reproduces the GIV of the Pleistocene. Our model is an efficient and spatially adimensional climate-cryosphere model with explicitly-solved physical processes that can be enabled or disabled independently, allowing us

to isolate the physical processes that control the mechanisms underlying the Pleistocene records. We describe the model in Sect. 2, together with results from each model configuration. In Sect. 3 we discuss the main results. Finally, the main conclusions are summarized in Sect. 4.

## 2 Model description and results

Here we describe the Physical Adimensional Climate-Cryosphere mOdel (PACCO) in a progressive manner in order to provide

a full picture of its capacities together with the physical basis for its formulation. PACCO represents conceptually the interaction between climate and Northern Hemisphere ice sheets using a system of up to 6 coupled ODEs (Ordinary Differential Equations) as described in the following sections. The experiments summarized in Table 1 progressively capture more processes and will therefore be used as a convenient way to describe the physics captured by PACCO.



The equations governing ice sheet dynamics are the same in all experiments. Thus, we first introduce the ice sheet thickness

$H$ evolution as

$$\frac{dH}{dt} = \dot{m} - q, \tag{1}$$

where $\dot{m}$ is the surface mass balance and $q$ is the ice discharge. The former will be described in Sect. 2.1 and the latter is

$$q = \boldsymbol{v} \cdot \boldsymbol{\nabla} H \tag{2}$$

where $\boldsymbol{v}$ is the ice-sheet velocity field. Since PACCO is a spatially dimensionless model, we employ an ice column approx-

imation with scalar velocity $v$ and thickness $H$. By using the typical length $L$ of the ice sheet, the spatial derivative can be

approximated (Oerlemans et al., 2008):

$$q = v \cdot \frac{H}{L}. \tag{3}$$

Ice velocity is decomposed into a deformation and a sliding component, respectively $v_d$ and $v_b$:

$$v = v_d + v_b. \tag{4}$$

The deformational velocity takes the form of Glen's flow law (Glen, 1958):

$$\boldsymbol{v}_d = \frac{2A_f}{n+2} \cdot H \cdot |\tau_d|^{n-1} \cdot \boldsymbol{\tau}_d, \tag{5}$$

with the usually used exponent $n = 3$ and $A_f$ the Glen's law flow parameter that represents the effect of the ice viscosity. In its

zero-dimensional form, Eq. (5) becomes

$$v_d = \frac{2}{5} \cdot A_f \cdot H \cdot \tau_d^3. \tag{6}$$

The basal velocity field is assumed to follow a Weertman-like sliding law (Cuffey and Paterson, 2010; Pattyn, 2010; Pollard

and DeConto, 2012),

$$\boldsymbol{v}_b = C' \cdot |\tau_b| \cdot \boldsymbol{\tau}_b, \tag{7}$$

that in its zero-dimensional version is

$$v_b = f_{\text{str}} \cdot C_s \cdot \tau_b^2, \tag{8}$$

where $C_s$ is a model parameter that represents the "raw" sliding coefficient derived from Pollard and DeConto (2012) and $f_{\text{str}}$

is a model parameter that represents the fraction of ice streams in the ice sheet. In Eqs. (6) and (8), the driving and basal stress

(respectively $\tau_d$ and $\tau_b$) are equal, following the Shallow Ice Approximation (SIA, Fowler and Larson, 1980; Hutter et al.,

1981), thus implying that the ice velocity does not surge but evolves smoothly. The driving stress that causes ice to deform and

move under its own weight, normally expressed as

$\boldsymbol{\tau}_d = \rho_{\text{ice}} \cdot g \cdot H \cdot \boldsymbol{\nabla} z, \tag{9}$





**Table 1.** Summary of the experiments performed in this work. Experiments are ordered in gradual increasing complexity.

| Experiment | Description | Section |
|---|---|---|
| LIN | SIF, ISD, linear SMB | 2.1 |
| NONLIN | SIF, ISD, non-linear SMB | 2.2 |
| ISOS | SIF, ISD, isostatic response | 2.3 |
| RISOS | RIF, ISD, isostatic response | 2.4 |
| BASE | RIF, ISD, carbon cycle, albedo | 2.5 |
| THERM | RIF, ISD, carbon cycle, albedo, IST | 2.6 |
| AGING | RIF, ISD, carbon cycle, albedo aging | 2.7 |

SIF = Synthetic Insolation Forcing, RIF = Real Insolation Forcing, ISD = Ice Sheet
Dynamics, IST = Ice Sheet Thermodynamics, SMB = Surface Mass Balance

is transformed to

$$\tau_d = \rho_{\text{ice}} \cdot g \cdot H \cdot \frac{z}{L}, \tag{10}$$

where $\rho_{\text{ice}}$ is the ice density ($\sim 910 \,\text{kg} \cdot \text{m}^{-3}$), $g$ is the gravitational acceleration ($9.81 \,\text{m} \cdot \text{s}^{-2}$) and $z$ the ice surface elevation. Finally, the ice-sheet surface elevation $z$ appearing in Eqs. (9) and (10) is given by

$$z = H + B, \tag{11}$$

with $B$ the elevation of the bedrock that remains constant for the moment.

### 2.1 A quasi linear configuration for surface mass balance (LIN experiment)

We start by building a simple configuration that represents the interaction between insolation and ice-sheet dynamics. The simplest configuration of PACCO receives insolation as the only forcing and, for that purpose, we defined a synthetic insolation forcing as a linear combination of three cosines with periods identical to those of orbital parameters (synthetic forcing and linear mass balance, LIN experiment, Table 1). Thus

$$I = I_{\text{ref}} + A_I \cdot \sum_{i=p,o,e} P_i \cdot cos\left(\frac{2\pi t}{\tau_i}\right), \tag{12}$$

where $I_{\text{ref}}$ is a reference value for insolation and $P_i$ and $\tau_i$ are the power and period associated to precession ($p$), obliquity ($o$) and eccentricity ($e$), respectively. $A_I = (I_{\text{max}} - I_{\text{min}}) \cdot 2^{-1}$ is the amplitude of the signal, with $I_{\text{min}}$ and $I_{\text{max}}$ parameters based on the minimum and maximum values of the real boreal summer solstice insolation at 65ºN (based on Laskar et al., 2004, after Berger, 1978). All parameter values employed are shown in Table 2. Eq. 12 thus provides a synthetic insolation forcing for the model. Insolation is then translated to sea-level temperature via

$$T_{\text{sl}} = T_{\text{ref}} + A_T \cdot \bar{I}, \tag{13}$$



**Table 2.** Parameters from LIN, NONLIN, ISOS and RISOS experiments. Note that the parameters not referenced correspond to model calibration values.

| Parameter | Value | Parameter | Value | Reference |
|---|---|---|---|---|
| $t_0$ (kyr BP) | 800 | $I_{\mathrm{ref}}$ (W $\cdot$ m$^{-2}$) | 480 | Present-day anomaly |
| $t_f$ (kyr BP) | 0 (J2000) | $P_p, P_o, P_e$ | 0.6, 0.4, 0.0 | Paleoclimatic constraint |
| $T_0$ ($^o$C) | 0 | $\tau_p, \tau_o, \tau_e$ (kyr) | 23, 41, 100 | Paleoclimatic constraint |
| $H_0$ (m) | 0 | $T_{\mathrm{ref}}$ ($^o$C) | 0 | Present-day anomaly |
| $B_0$ (m) | 500 | $A_T$ (K) | 15 | Paleoclimatic constraint |
| $\rho_{\mathrm{bed}}$ (kg $\cdot$ m$^{-3}$) | 2700 | $I_{\min}$ (W $\cdot$ m$^{-2}$) | 425 | Paleoclimatic constraint |
| $\rho_{\mathrm{ice}}$ (kg $\cdot$ m$^{-3}$) | 910 | $I_{\max}$ (W $\cdot$ m$^{-2}$) | 565 | Paleoclimatic constraint |
| $\lambda$ (m$\cdot$yr$^{-1}\cdot$K$^{-1}$) | 0.2 | $L$ (km) | 1000 | Oerlemans et al. (2008) |
| $T_{\mathrm{thr}}$ ($^o$C) | -5 | $f_{\mathrm{str}}$ | 0.2 | Margold et al. (2015) |
| $\dot{s}_{\mathrm{ref}}$ (m $\cdot$ yr$^{-1}$) | 0.2 | $A_f$ (Pa$^{-3}\cdot$yr$^{-1}$) | $10^{-16}$ | Glen (1958) |
| $k_{\dot{s}}$ (m $\cdot$ yr$^{-1} \cdot$ K$^{-1}$) | 0.0015 | $C_s$ (m$\cdot$yr$^{-1}\cdot$Pa$^{-2}$) | $10^{-10}$ to $10^{-4}$ | Pollard and DeConto (2012) |
| $B_{\mathrm{ref}}$ (m) | 500 | $\tau_B$ (yr) | 5000 | Le Meur and Huybrechts (1996); |
| | | | | Swierczek-Jereczek et al. (2023) |

where $T_{\mathrm{ref}}$ is a reference value for the Earth's temperature, $A_T$ is the amplitude of the signal and

$$\bar{I} = 2 \cdot \frac{I - I_{\min}}{I_{\max} - I_{\min}} - 1, \tag{14}$$

that is, the normalized insolation between -1 and 1. In this way we have built an extremely simple climate model. The "climate response" is fed into the ice sheet through the surface mass balance $\dot{m}$. This is assumed to be the only source of mass balance of the ice sheet (i.e, basal mass balance and calving are ignored) with the exception of the ice discharge (Eq. 3). Thus,

$$\dot{m} = \dot{s} - \dot{a}, \tag{15}$$

where the dot indicates rate of ice mass change (m $\cdot$ yr$^{-1}$); hence $\dot{s}$ is snowfall and $\dot{a}$ is ablation. Snowfall evolves linearly with the anomaly in temperature relative to a reference value $T_{\mathrm{ref}}$; this represents a linearization of the Clausius-Clapeyron equation:

$$\dot{s} = \dot{s}_{\mathrm{ref}} + k_{\dot{s}} \cdot (T_{\mathrm{sl}} - T_{\mathrm{ref}}), \tag{16}$$

with $\dot{s}_{\mathrm{ref}}$ and $k_{\dot{s}}$ model parameters (c.f. Table 2). The ablation term $\dot{a}$ in Eq. (15) follows a similar approach to the Positive Degree Day method (Braithwaite, 1980; Reeh, 1991; Ritz et al., 1996; Cuffey and Marshall, 2000; Huybrechts et al., 2004; Charbit et al., 2008; Robinson et al., 2010) that depends on the number of days in a year with sea-level temperature above the melting point. PACCO assumes a linear relationship between the ablation and the temperature anomaly and reduces the melting point constraint to $T_{\mathrm{thr}}$. In this way, we can allow for melting at lower temperatures to account for the lack of spatial





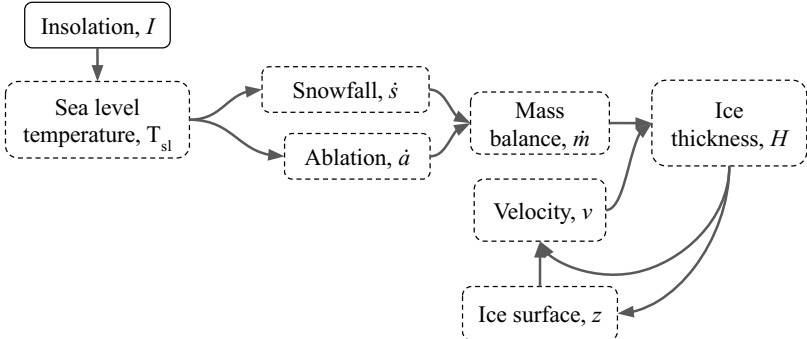

**Figure 2.** LIN experiment scheme. Note the linear relationship between the forcing and the evolution of the ice thickness.

and temporal knowledge. We then define ablation as

$$\dot{a} = \lambda \cdot (T_{\mathrm{sl}} - T_{\mathrm{thr}}). \tag{17}$$

Here, $\lambda$ is a parameter that transfers temperature anomaly to ice mass loss. Note that both $\dot{s}$ and $\dot{a}$ are defined as strictly positive. The model structure and flow chart are described in Fig. 2. This configuration essentially consists of a linear relation between insolation and mass balance and a nonlinear one between ice thickness and velocity. As expected, when integrating over 800 kyr, glacial inceptions and deglaciations respectively occur shortly after insolation minima and maxima with a pattern that resembles the sawtooth observed in proxy records (Fig. 3). Interestingly, the power spectrum of the response of $H$ and $v$ presents a peak around periodicities of 60 kyr, which is however absent from the forcing. This is the manifestation of the nonlinearities introduced by the ice dynamics and is the consequence of the fact that certain interglacials are very long, since the system evolves explicitly with insolation and only those insolation minima below a certain threshold allow for a positive mass balance, leading to an increase of the ice mass. In short, the ice-sheet dynamics creates a certain degree of nonlinearity that appears, however, to be not powerful enough to allow for a realistic GIV simulation.

## 2.2 Introducing feedbacks on surface mass balance (NONLIN experiment)

In order to build a more physically motivated model, important feedbacks must be included. The ice-surface elevation feedback (Weertman, 1961; Clark and Pollard, 1998; Oerlemans, 2003) is known to be a fundamental process controlling ice-sheet accumulation and ablation. The elevation feedback accelerates melting under a shrinking ice sheet and limits ablation when the ice sheet is growing, leading to hysteresis in the ice sheet's volume with respect to temperature (Robinson et al., 2012; Garbe et al., 2020). To take this feedback into account, the mass balance equations are modified by using $T_{\mathrm{surf}}$ instead of $T_{\mathrm{sl}}$, where

$$T_{\mathrm{surf}} = T_{\mathrm{sl}} - \Gamma \cdot z \tag{18}$$



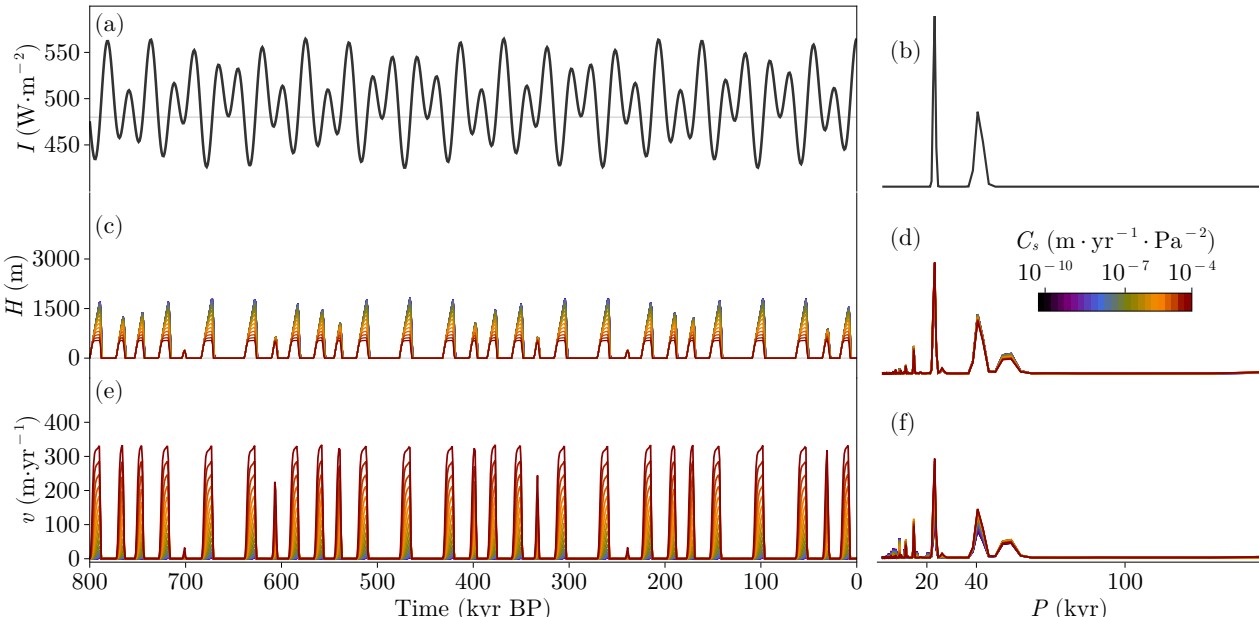

**Figure 3.** Results of the LIN simulation. (a, c, e) Time series obtained from the model using different sliding factors $C_s$. (b, d, f) Periodograms obtained from the time series in the left column. Note that when normalizing spectra, series were cut off for periods larger than 200 kyr.

and

$$\dot{s} = \dot{s}_{\text{ref}} + k_{\dot{s}} \cdot (T_{\text{surf}} - T_{\text{ref}}), \tag{19}$$

$$\dot{a} = \lambda \cdot (T_{\text{surf}} - T_{\text{thr}}), \tag{20}$$

where $\Gamma = 0.0065 \ \text{K} \cdot \text{m}^{-1}$ is the atmospheric lapse rate. Introducing the elevation feedback in the mass balance (Fig. 4) clearly alters the response to the forcing (Fig. 5). If the ice sheet is dynamic enough (i.e. high values of the sliding factor $C_s$), the model resonates to certain multiples of the insolation fundamental periods. This means that the elevation feedback introduces a non linearity via the modulation of the height amplitude, which increases the inertia of the system (Abe-Ouchi et al., 2013). Therefore, as in the previous formulation (LIN), periodicities of 60 kyr can be found, but also 80, 100 and 120 kyrs peaks now emerge. Still, this phenomenon does not produce a dominant GIV periodicity of 100 kyr.

## 2.3 Delayed isostatic adjustment (ISOS experiment)

Changes in ice load lead to delayed vertical bedrock motion, a process commonly known as isostatic adjustment. This effect is included here via another prognostic variable of the model (Fig. 6), such that the bedrock elevation $B$ evolves according to

$$\frac{dB}{dt} = \frac{\left[B_{\text{ref}} - \frac{\rho_{\text{ice}}}{\rho_{\text{bed}}} \cdot H\right] - B}{\tau_B}, \tag{21}$$



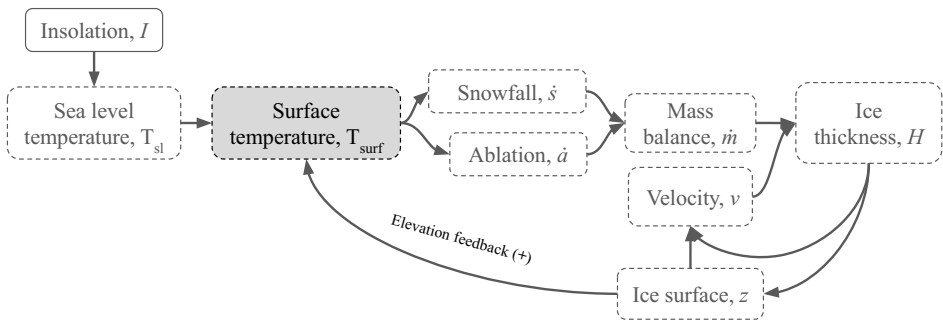

**Figure 4.** NONLIN experiment scheme. The previously linear relationship between the forcing and the evolution of the ice thickness (LIN) is now nonlinear due to the elevation feedback. Model additions with respect to LIN are highlighted.

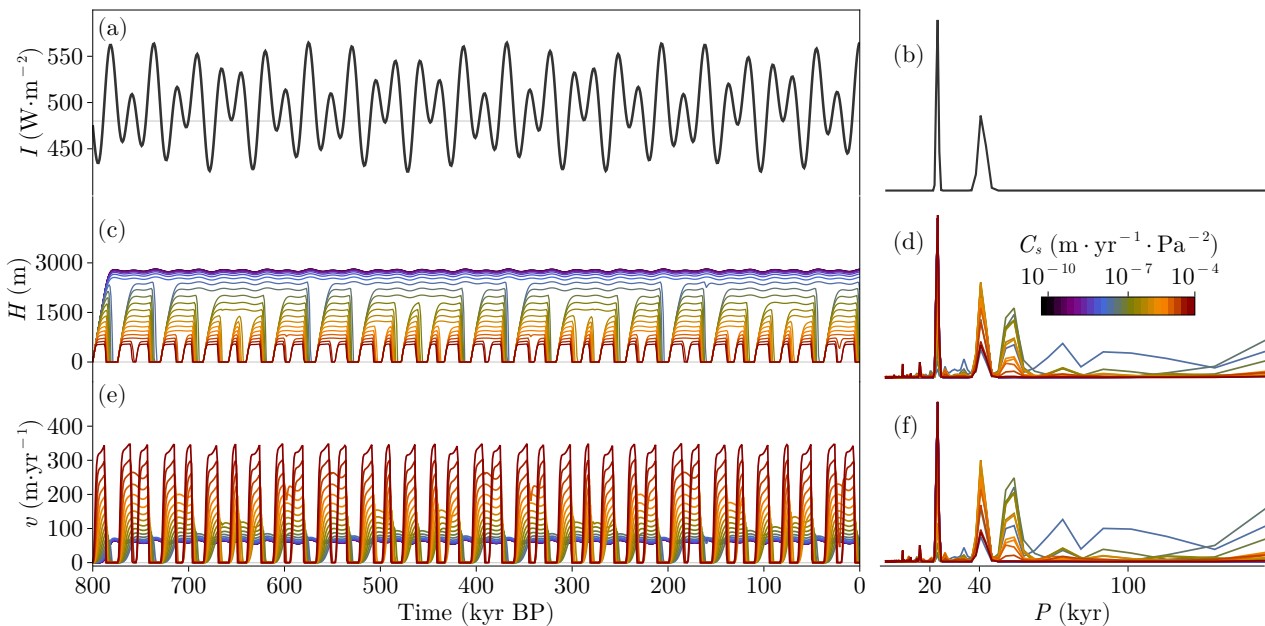

**Figure 5.** Results of the NONLIN simulation. (a, c, e) Time series obtained from the model using different sliding factors $C_s$ and (b, d, f) associated periodograms. Note that when normalizing spectra, series were cut off for periods larger than 200 kyr.

where $B_{\mathrm{ref}}$ is a model parameter representing the elevation of Earth's surface in the absence of ice, $\rho_{\mathrm{bed}}$ is the bedrock density (c.f. Table 2), and $\tau_B$ is the relaxation time of the bed. Fig. 7 shows, as in previous cases, the results of a simulation for different levels of the sliding factor. Resonance of the system to longer periods is favored for the less dynamic runs (lowest $C_s$ values). Reaching a glacial termination is now facilitated by the fact that ablation is greater than accumulation at low elevations (Fig. 8). We can define an elevation threshold $z_{\mathrm{thr}}$ such that when $z \leq z_{\mathrm{thr}}$ (i.e. $z$ is located below the red curve in Fig. 8) ablation





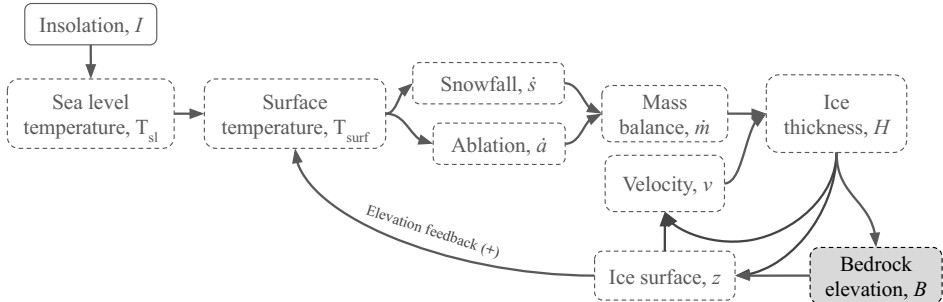

**Figure 6.** ISOS experiment scheme. Note that the surface elevation feedback is altered due to bedrock response. Model additions with respect to NONLIN are highlighted.

is surpassed by accumulation. In other words, $z_{\text{thr}}$ represents the equilibrium line altitude in the ice sheet:

$$z_{\text{thr}} = \frac{\dot{s}_{ref} + k_{\dot{s}} \cdot (T_{sl} - T_{ref}) - \lambda \cdot (T_{sl} - T_{thr})}{(k_{\dot{s}} - \lambda) \cdot \Gamma}. \tag{22}$$

This happens more easily for the ISOS experiments since the simulated ice-sheet heights are generally lower for the same set of parameters, as a consequence of the bedrock depression induced by the ice load. Thus, isostasy favors glacial terminations

and 100 kyrs cycles can now more clearly emerge for certain values of the sliding factor. We must bear in mind, however, that so far our insolation forcing is synthetic. We therefore next turn to a more realistic forcing to assess whether this formulation allows for a good match with paleodata.

## 2.4  ISOS configuration with real insolation (RISOS experiment)

To answer if the only missing piece is the form of the forcing we perform the RISOS experiment with the ISOS configuration

(Fig. 6) but using the boreal summer solstice insolation at 65ºN (Fig. 9) as forcing $I$ (obtained following Berger, 1978; Laskar et al., 2004). The model produces resonance at higher periodicities than the ones corresponding to obliquity and precession, so glacial terminations occur at different multiples of the former, depending on the sliding intensity (Fig. 9). However, the simulated timing does not match yet that provided by paleoclimatic proxies. The model response is still quite linear with the insolation forcing, so the amplitude of some of its peaks is not enough to make the ice sheet deglaciate (for low sliding) or,

on the contrary, a moderate increase in insolation easily induces a termination (for high sliding). The absence of a satisfactory synchronization with the observed deglaciations, even for mid-range values of the sliding factor, likely indicates that the model still lacks some important climatic processes. These will be addressed in the next section.

## 2.5  Improving the coupling between ice sheet and climate (BASE experiment)

The first required improvement concerns the treatment of air temperature in the model, which is now regionalized in order to

include the two-way interaction between the atmosphere and the cryosphere, as well as its response to the radiative forcing.





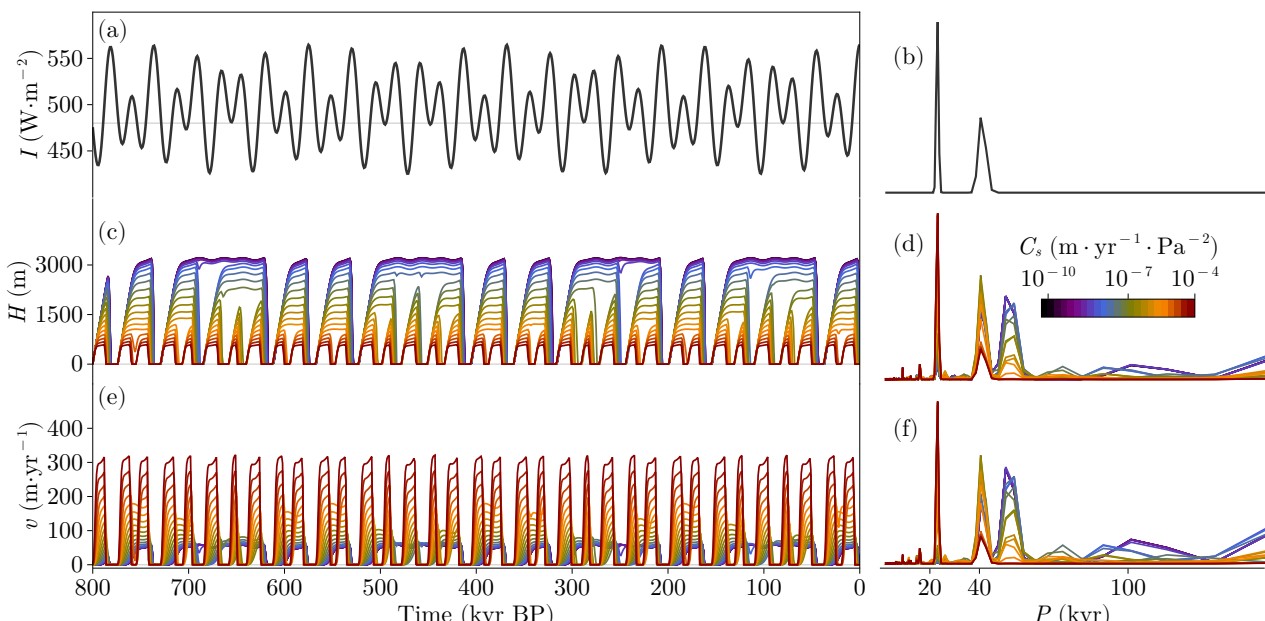

**Figure 7.** Results from the ISOS simulation. (a, c, e) Time series obtained from the model using different sliding factors $C_s$ and (b, d, f) associated periodograms. Note that when normalizing spectra, series were cut off for periods larger than 200 kyr.

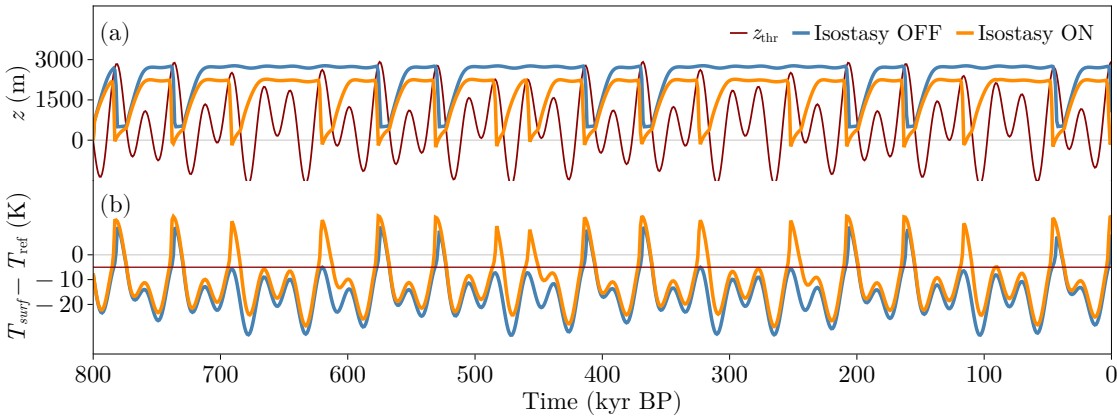

**Figure 8.** Time series for (a) ice surface elevation and $z_{\mathrm{thr}}$ and (b) surface temperature anomaly (with respect $T_{\mathrm{ref}}$). $z_{\mathrm{thr}}$ is the point where $\dot{a}$ surpasses $\dot{s}$, and the horizontal red line indicates $T_{\mathrm{thr}}$ which is the isoline where $\dot{a}$ becomes positive. Note that the plotted curves are for the same sliding factor $C_s = 5.8 \cdot 10^{-8}\,\mathrm{m} \cdot \mathrm{yr}^{-1}\mathrm{Pa}^{-2}$.

The regional air temperature $T$ hence evolves in time as follows:

$$\frac{dT}{dt} = \frac{[T_{\mathrm{ref}} + c_I \cdot R_I + c_C \cdot R_C - c_z \cdot z] - T}{\tau_T} \tag{23}$$





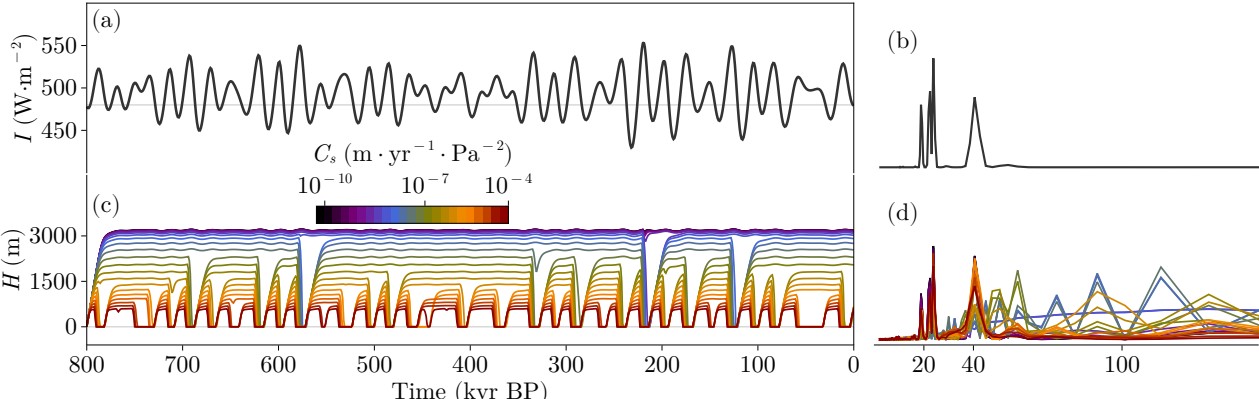

**Figure 9.** Results of the RISOS simulation. (a, c, e) Time series obtained from the model using different sliding factors $C_s$. (b, d, f) Periodograms obtained from the time series in the left column. Note that when normalizing spectra, series were cut off for periods larger than 200 kyr.

with $c_I, c_C, c_Z$ the climate sensitivities to insolation, atmospheric $CO_2$ radiative forcing and ice-sheet height, respectively, and $\tau_T$ the thermal characteristic time. In this way, positive anomalies in insolation or atmospheric $CO_2$ concentration $C$ tend to
increase temperature, while the ice-sheet size tends to decrease it. Note that the local effect of $T_{\mathrm{surf}}$ is now translated to a more regional effect via $T$, which is affected by the carbon cycle and the size of the northern hemisphere ice sheets. Finally, $R_I$ and $R_C$ are the radiative forcing associated with $I$ and $C$, respectively defined as

$$R_I = I - I_{\mathrm{ref}} \tag{24}$$

and

$$R_C = 5.35 \cdot log\left(\frac{C}{C_{\mathrm{ref}}}\right). \tag{25}$$

The latter was formulated by Myhre et al. (1998) and it is commonly employed in conceptual modeling. Here $C_{\mathrm{ref}}$ is a reference value for carbon dioxide (typically set to the pre-industrial value of 280 ppm) and

$$\frac{dC}{dt} = \frac{[C_{\mathrm{ref}} + k_{T,C} \cdot (T - T_{\mathrm{ref}})] - C}{\tau_C}. \tag{26}$$

This reflects the capability of the sources and sinks of atmospheric $CO_2$ to vary with temperature.

Another improvement in the model is the ablation term, that now follows an ITM-like parameterization (Insolation Temperature Melting method, Pellicciotti et al., 2005; Van Den Berg et al., 2008; Robinson et al., 2010)

$$\dot{a} = k_I \cdot (1 - \alpha) \cdot (I - I_{\mathrm{ref}}) + \lambda \cdot (T - T_{\mathrm{thr}}) \tag{27}$$

where $k_I$ is the sensitivity of ablation to shortwave radiation (insolation). This improvement was made in order to take into account both short and long wave radiation. As mentioned before, since ice surface elevation is already included in Eq. (23),





**Table 3.** Parameters from BASE, THERM and AGING experiments. Note that the parameters not referenced correspond to model calibration values.

| Parameter | Value | Parameter | Value | Reference |
|---|---|---|---|---|
| $t_0$ (kyr BP) | 900 | $T_{\mathrm{ref}}$ ($^o$C) | 0 | Present-day anomaly |
| $t_f$ (kyr BP) | 0 (J2000) | $S_{\mathrm{ocn}}$ (km$^2$) | $3.618 \cdot 10^8$ | Cogley (2012) |
| $T_0$ ($^o$C) | 0 | $\alpha_l, \alpha_{\mathrm{ni}}, \alpha_{\mathrm{oi}}$ | 0.2, 0.9, 0.25 | Cuffey and Paterson (2010) |
| $H_0$ (m) | 0 | $I_{\mathrm{ref}}$ (W$\cdot$m$^{-2}$) | 480 | Present-day anomaly |
| $B_0$ (m) | 500 | $L$ (km) | 1000 | Oerlemans et al. (2008) |
| $C_0$ (ppm) | 280 | $C_{\mathrm{ref}}$ (ppm) | 280 | Present-day anomaly |
| $A_0$ (yr) | 0 | $C_s$ (m·yr$^{-1}$·Pa$^{-2}$) | $10^{-10}$ to $10^{-4}$ | Pollard and DeConto (2012) |
| $f_{str,0}$ | 0.0 | $A_{\mathrm{thr}}(K)$ | 20 | Paleoclimatic constraint |
| $T_{ice,0}$ ($^o$C) | -15 | $f_{\mathrm{str}}$ | 0.2 | Margold et al. (2015) |
| $\rho_{\mathrm{bed}}$ (kg $\cdot$ m$^{-3}$) | 2700 | $k_I$ (m $\cdot$ yr$^{-1}\cdot$ W$^{-1}\cdot$ m$^{-2}$) | 0.027 | |
| $\rho_{\mathrm{ice}}$ (kg $\cdot$ m$^{-3}$) | 910 | $\lambda$ (m·yr$^{-1}$·K$^{-1}$) | 0.1 | |
| $\rho_{\mathrm{wtr}}$ (kg $\cdot$ m$^{-3}$) | 1000 | $T_{\mathrm{thr}}$ ($^o$C) | -5 | |
| $\tau_T$ (yr) | 900 | $\dot{s}_{\mathrm{ref}}$ (m $\cdot$ yr$^{-1}$) | 0.3 | |
| $k_T$ (J $\cdot$ s$^{-1}\cdot$ m$^{-1}\cdot$ K$^{-1}$) | 2.1 | $k_{\dot{s}}$ (m $\cdot$ yr$^{-1}\cdot$ K$^{-1}$) | 0.0015 | |
| $c_I$ (K $\cdot$ m$^2\cdot$ W$^{-1}$) | 0.1 | $\nu$ (m·yr$^{-1}$) | 300 | |
| $c_C$ (K $\cdot$ m$^2\cdot$ W$^{-1}$) | 0.65 | $A_f$ (Pa$^{-3}$·yr$^{-1}$) | $10^{-16}$ | Glen (1958) |
| $c_z$ (K $\cdot$ m$^{-1}$) | 0.007 | $B_{\mathrm{ref}}$ (m) | 500 | |
| $k_{T,C}$ (ppm $\cdot$ K$^{-1}$) | 5 | $H_b$ (m) | 10 | Robel et al. (2013) |
| $h_{geo}$ (W·m$^{-2}$) | $10^{-3}$ to $10^{-2}$ | $\tau_{kin}$ (yr) | 10 to $10^4$ | Payne et al. (2004); Nick et al. (2009) |
| $\tau_C$ (yr) | 10 | $f_{\mathrm{str,min}}$ | 0.0 | |
| $f_{\mathrm{str,max}}$ | 0.5 | $\tau_\alpha$ (yr) | $10^3$ to $10^6$ | |
| $k_\alpha$ (yr$^{-1}$) | $5 \cdot 10^{-6}$ | $\tau_B$ (yr) | 5000 | Le Meur and Huybrechts (1996); Swierczek-Jereczek et al. (2023) |

ablation and snowfall use $T$ instead of $T_{\mathrm{surf}}$:

$$\dot{s} = \dot{s}_{\mathrm{ref}} + k_{\dot{s}} \cdot (T - T_{\mathrm{ref}}).  \tag{28}$$

This model configuration is now represented by Fig. 10 and the results of sensitivity experiments to different $C_s$ values are shown in Fig. 11. The sliding strength modulates the amplitude of the ice thickness and thus, the ice-sheet sensitivity to full deglaciation. With this, we produce a more realistic 100 kyr periodicity in $H$ than in RISOS when also using real insolation
forcing. However, the timing and periodicity of $T$ are not satisfactory yet, suggesting the nonlinear response of the model is still too weak to produce reliable GIV.





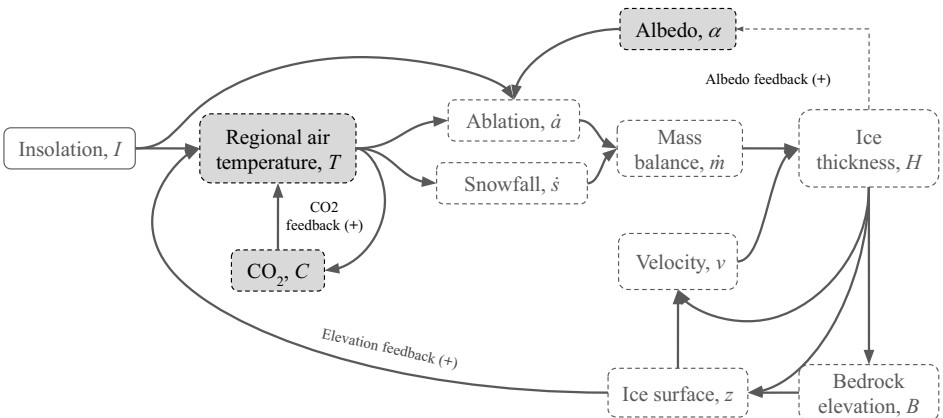

**Figure 10.** BASE experiment scheme. Model additions with respect to ISOS are highlighted.

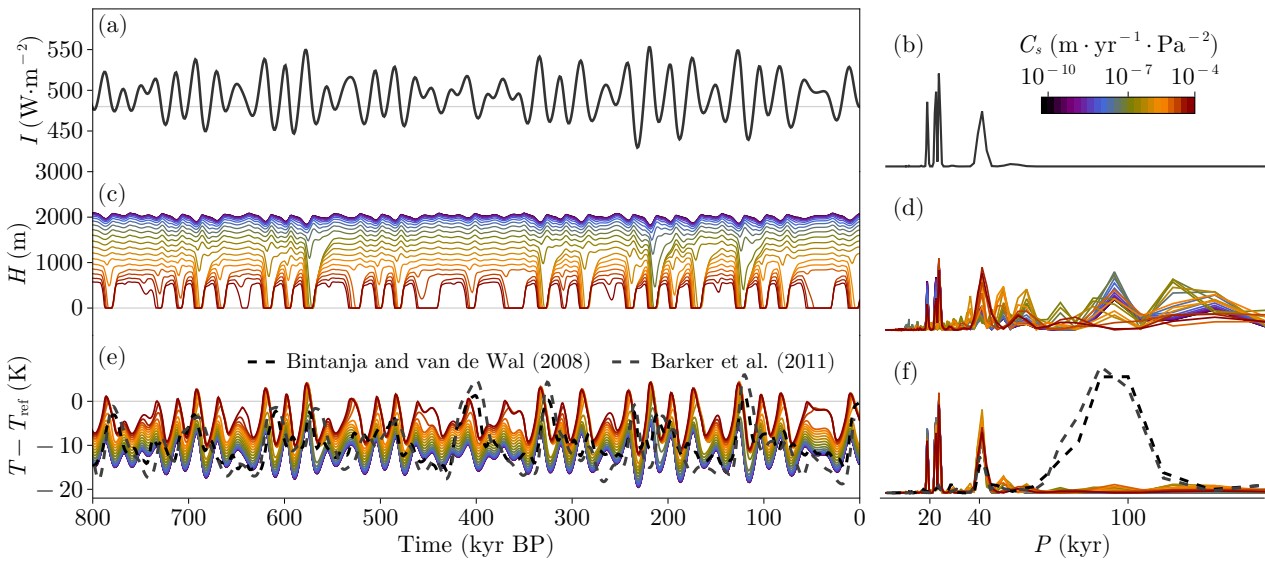

**Figure 11.** Results of the BASE simulation. (a, c, e) Time series obtained from the model using different sliding factors $C_s$. (b, d, f) Periodograms obtained from the time series in the left column. Note that when normalizing spectra, series were cut off for periods larger than 200 kyr. Note that in (e, f) dashed lines refer to two different proxies.

Based on the literature, there are at least two other ways (Ganopolski, 2024) to amplify the model's nonlinear behavior: through ice-sheet thermodynamics and through ice darkening, which will both be investigated in the next section as possible sources of improved GIV accuracy.



### 2.6 Ice-sheet thermodynamics (THERM experiment)

One way to enhance the nonlinear response of the system is via ice-sheet thermodynamics and its effect on the basal sliding and streaming potential of the ice sheet (i.e. thermodynamic hypothesis, Verbitsky et al., 2018; Ganopolski, 2024). The temperature of the base of an ice sheet ($T_{ice}$) is influenced by both ice insulation and ice creep. If basal temperature reaches the pressure melting point, basal sliding is enhanced and ice streams accelerate and expand further. To parameterize this process we consider the interaction between a cold and a temperate environment, the air and the bed, respectively. At the base

$$dQ = \rho_{\mathrm{ice}} \cdot S \cdot H_b \cdot c_{\mathrm{ice}} \cdot dT_{\mathrm{ice}}, \tag{29}$$

with $c_{\mathrm{ice}} = 2009 \ \mathrm{J \cdot Kg^{-1} \cdot K^{-1}}$ the ice specific heat capacity, $S$ the ice surface and $H_b$ the temperate base thickness (Robel et al., 2013). If we divide by the timestep and $S$ we get an equation for the evolution in time of the basal ice temperature $T_{\mathrm{ice}}$:

$$\frac{dT_{\mathrm{ice}}}{dt} = \frac{1}{c_{\mathrm{ice}} \cdot \rho_{\mathrm{ice}} \cdot H_b} \cdot \frac{dh}{dt}, \tag{30}$$

with

$$h = Q/S = \sum_i h_i \tag{31}$$

that is the sum of the different heat fluxes between the temperate and cold environments. Now, we will define the different components. The bed of an ice sheet (temperate environment) is exposed to the geothermal heat flow $h_{\mathrm{geo}}$ from the lithosphere, here taken to be a constant parameter (Table 3), and, to the heat flux due to basal drag:

$$h_{\mathrm{drag}} = v_b \cdot \tau_b. \tag{32}$$

These two terms act as a heat source to the ice sheet through its base. At the air-ice interface (cold environment) there is a heat loss due to ice conduction ($h_{\mathrm{cond}}$):

$$h_{\mathrm{cond}} = -k_T \cdot \nabla T_{\mathrm{ice}}, \tag{33}$$

with $k_T$ the ice thermal conductivity (Table 3). Note that we have neglected advection in both the horizontal and the vertical dimensions since the former is assumed to be zero when treating the entire ice sheet and the latter is negligible in the heat balance ($\sim 10^{-6} \ \mathrm{W \cdot m^{-2}}$). Eq. (33) relies on the assumption that conduction can be defined across the ice column (assumed to be an isotropic medium) as:

$$h_{\mathrm{cond}} = k_T \cdot \frac{T - T_{\mathrm{ice}}}{H}. \tag{34}$$

Once the thermodynamics are defined, the effect on ice-sheet dynamics is translated through the fraction of ice streams $f_{\mathrm{str}}$ within the ice sheet, which evolves in time according to

$$\frac{df_{\mathrm{str}}}{dt} = \frac{f_{\mathrm{str,ref}} - f_{\mathrm{str}}}{\tau_{\mathrm{kin}}}, \tag{35}$$





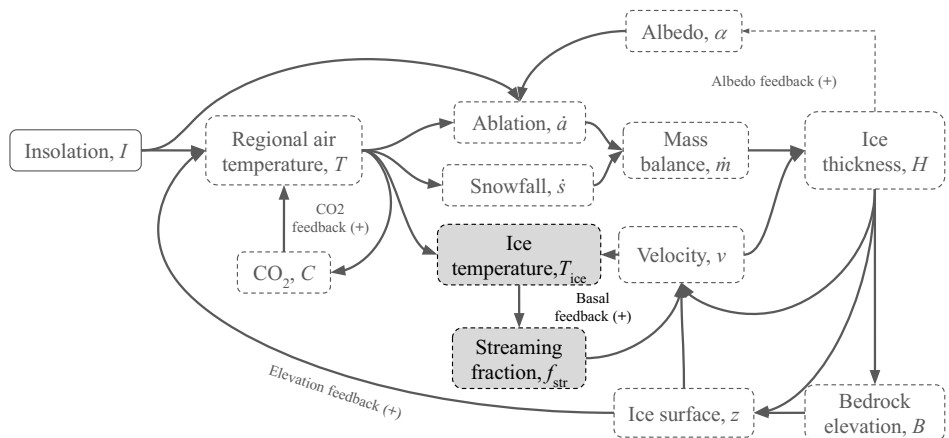

**Figure 12.** THERM experiment scheme. Model additions with respect to BASE are highlighted.

with $\tau_{\mathrm{kin}} = L \cdot v_{\mathrm{kin}}^{-1}$ the time that an ice stream needs to propagate into the interior of the ice sheet (Nye, 1963; Jóhannesson et al., 1989; Payne et al., 2004; Nick et al., 2009). In this way, we account for the fact that the entire ice-sheet base does not become temperate at once, but rather gradually with a characteristic time $\tau_{\mathrm{kin}}$. The reference value of streaming $f_{\mathrm{str,ref}}$ depends
explicitly on the thermodynamics:

$$f_{\mathrm{str,ref}} = (f_{\mathrm{str,max}} - f_{\mathrm{str,min}}) \cdot p_c + f_{\mathrm{str,min}}, \tag{36}$$

where $f_{\mathrm{str,max}}$ and $f_{\mathrm{str,min}}$ are model parameters based on glaciological constraints, and $p_c$ is the propagation coefficient

$$p_c = \frac{T_{\mathrm{ice}} - T_{\mathrm{str}}}{T_{\mathrm{mp}} - T_{\mathrm{str}}}. \tag{37}$$

Here, $T_{\mathrm{mp}}$ is the ice melting point temperature (0 °C) and $T_{\mathrm{str}}$ the temperature boundary that allows streaming propagation; $p_c$ is
a number between 0 and 1 that accounts for the state of the base. If $T_{\mathrm{ice}} \leq T_{\mathrm{str}}$, $p_c = 0$ and $f_{\mathrm{str,ref}} = f_{\mathrm{str,min}}$ are imposed since the base is frozen. If, on the contrary, $T_{\mathrm{ice}} > T_{\mathrm{str}}$, the base starts to be temperate. Thus, $0 < p_c \leq 1$ and $f_{\mathrm{str,min}} < f_{\mathrm{str,ref}} \leq f_{\mathrm{str,max}}$. At this point, the ice sheet will tend via Eq. (35) to present more streaming and basal sliding. We limit $p_c$ to 1 and $f_{\mathrm{str,ref}} = f_{\mathrm{str,max}}$.

 Sensitivity experiments with respect to $h_{\mathrm{geo}}$ and $\tau_{\mathrm{kin}}$ are carried out (Fig. 13). The former shows that if $h_{\mathrm{geo}}$ is very large,
the model cannot lose enough heat and the ice sheet experiences strong, frequent streaming, yielding typical periodicities below 60 kyr. Thus, $H$ is modulated in a similar fashion to the previous purely dynamic experiments. However, for a more balanced $h_{\mathrm{geo}}$ with the rest of the components, $T_{\mathrm{ice}}$ evolves more slowly and the streaming is also propagated more slowly, producing oscillations of about 80-100 kyr. In contrast, the response displays quite low sensitivity when sampling $\tau_{\mathrm{kin}}$ within a plausible range, $\tau_{\mathrm{kin}} \leq 1000$ yr, there are not many differences and the ice sheet tends to oscillate with a realistic GIV
amplitude but the timing is not satisfactory. Note that in these experiments $C_s$ was fixed to $10^{-5}$ m·yr$^{-1}$·Pa$^{-2}$ since a sensitivity experiment for this parameter (not shown) showed that the purely dynamic nature of the thermodynamic hypothesis



was in need of higher basal velocities. Despite the introduction of enhanced ice dynamics via the addition of basal velocities dependent on ice thermodynamics, the THERM configuration does not capture the 100 kyr periodicity well.

### 2.7 Ice aging (AGING experiment)

Another way of amplify the nonlinear response of the Earth system to insolation forcing is through the reduction in ice albedo due to its natural darkening and dust deposition (Ganopolski and Calov, 2011; Willeit et al., 2019; Ganopolski, 2024). The former is related to the compaction of snow and the latter due to the aridity of glacial landscapes, which favors the deposition of dust on ice sheets. These phenomena can be easily implemented in PACCO via a relaxation equation for the albedo $\alpha$:

$$\frac{d\alpha}{dt} = \frac{\alpha_{\mathrm{ref}} - \alpha}{\tau_\alpha}, \tag{38}$$

with $\tau_\alpha$ the relaxation time associated with the change in albedo for the entire ice sheet. The target value $\alpha_{\mathrm{ref}}$ changes with the age of the ice sheet. For simplicity, it follows a linear relationship

$$\alpha_{\mathrm{ref}} = \alpha_{\mathrm{ni}} - k_\alpha \cdot A, \tag{39}$$

where $k_\alpha$ is a parameter that defines the aging rate and $A$ is the ice age that depends on the presence or not of ice. The minimum value of $\alpha_{\mathrm{ref}}$ is $\alpha_{\mathrm{oi}}$, which is the estimated albedo value of old ice. If there is no ice we impose $\alpha = \alpha_{\mathrm{l}}$, which is the value of

land albedo. Now, albedo therefore evolves to a reference value that changes in time when there is ice. In this manner, if the ice is maintained enough in time, its decreasing albedo results in a nonlinear increase of ablation. However, the assumption that aging affects the entire ice sheet is a crude one, since the accumulation areas are covered by fresh snow and, therefore, their contribution to ablation must be smaller. Thus, we add an additional constraint

$$\alpha = \alpha_{\mathrm{ni}} \quad if \; \dot{m} > 0. \tag{40}$$

In this way, if snowfall outweighs ablation, we assume that ablation zones are covered by snow (snow/new ice albedo, $\alpha_{\mathrm{ni}}$), but if ablation is higher than snowfall, glacier ice is exposed. In this way, the mass balance sees the time-evolving value of albedo. The result is the model configuration represented in Fig. 14.

Sensitivity experiments with respect to the sliding factor $C_s$ show that a value from $10^{-10}$ to $10^{-7}$ (range from Pollard and DeConto, 2012) produces realistic oscillations (Fig. A1). Thus, we fixed the value of $C_s$ to $10^{-9}$ based on the idea that Late

Pleistocene ice sheets were slow and not quite dynamic (Berends et al., 2021a, , Fig. 15). Our results show that the aging time $\tau_\alpha$ is relevant to reproduce the correct GIV. In this experiment, 1000 yr seems to be a good value (highlighted in black in Fig. 15) since it provides high spectral power in 100 kyr band and a good agreement with proxies. Thus, we will compare the rest of the states of this model run with different proxies.

Since PACCO has no spatial dimensions, the ice volume is defined through the product of ice thickness and the potentially

glaciated surface $S$. This can be expressed in meters of sea level equivalent (m SLE) by

$$V_{ice} = \frac{\rho_{\mathrm{ice}}}{\rho_{\mathrm{wtr}} \cdot S_{\mathrm{ocn}}} \cdot S(T, v_b) \cdot H, \tag{41}$$







**Figure 13.** Results of the THERM simulation. (a, c, e, g) Time series obtained from the model using different sliding factors $C_s$. (i, k, m, o) Time series obtained from the model using different characteristic times, $\tau_{\text{kin}}$. (b, d, f, h, j, l, n, p) Periodograms obtained from the time series in the left column. Note that when normalizing spectra, series were cut off for periods larger than 200 kyr.



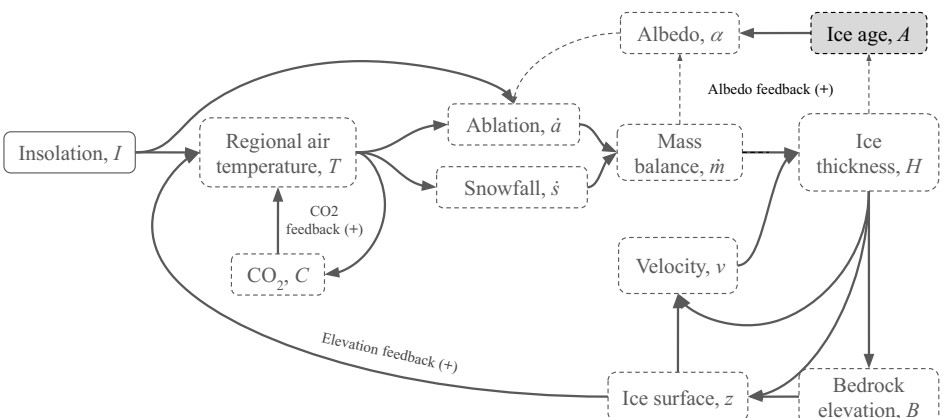

**Figure 14.** AGING experiment scheme. Model additions with respect to BASE are highlighted.

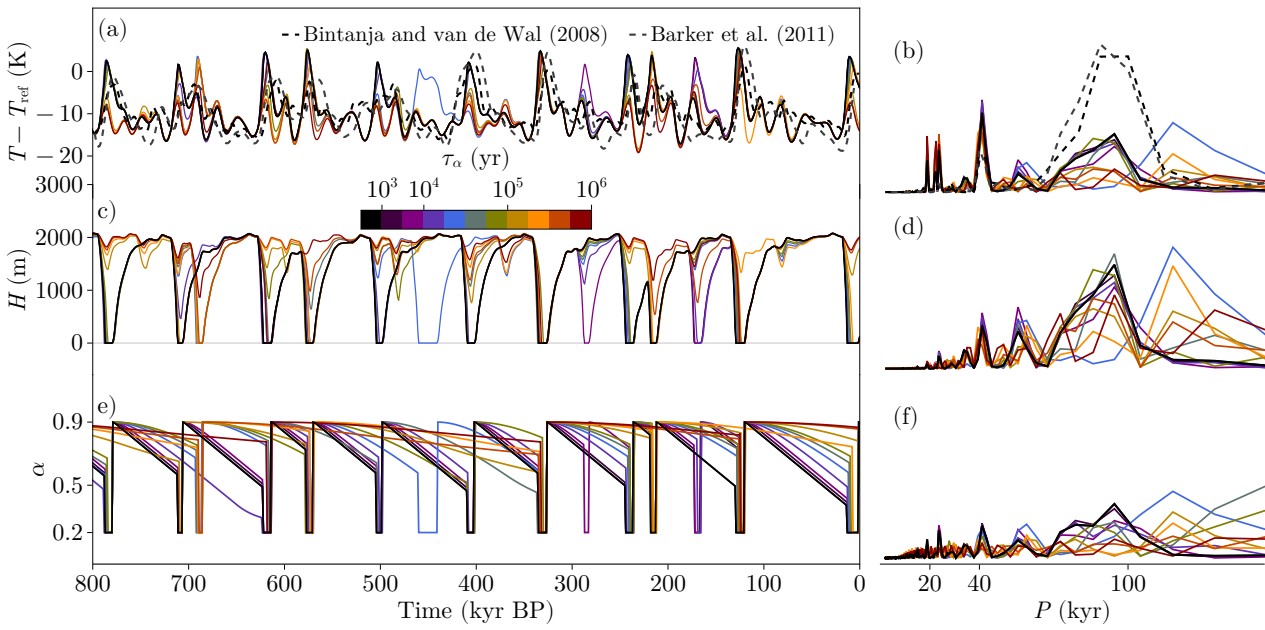

**Figure 15.** Results of the AGING simulation. (a, c, e) Time series obtained from the model using different aging times $\tau_\alpha$. (b, d, f) Periodograms obtained from the time series in the left column. Note that when normalizing spectra, series were cut off for periods larger than 200 kyr.

where $\rho_{\mathrm{wtr}}$ is water's density ($1000 \, \mathrm{kg \cdot m^{-3}}$), $S_{\mathrm{ocn}} (3.618 \cdot 10^8 \ \mathrm{km^2})$ is the oceanic surface of the Earth (Cogley, 2012) and

$$S = 8 \cdot \pi \cdot L^2 \cdot \left( \frac{T - T_{\mathrm{ref}}}{A_{\mathrm{thr}}} - \frac{v_b}{v} \right). \tag{42}$$





This equation relies on the fact that a theoretical ice sheet is a symmetrical dome ($\pi \cdot L^2$) whose extension can be modified by both anomalies in regional climate $(T - T_{\text{ref}}) \cdot A_{\text{thr}}^{-1}$ and excessive sliding velocities $v_b \cdot \upsilon^{-1}$. Note that $A_{\text{thr}}$ is the thermal amplitude provided by a certain extent of the ice sheet, $\upsilon$ is the typical velocity in an ice stream and $S$ is multiplied by 8 that is a free constant parameter to obtain similar amplitude to proxies.

Comparison of PACCO's results with proxy data (Fig. 16) shows that for the main variables of the model ($T$ and $H$), the periodograms exhibit greater power around 100 kyr while maintaining certain power in obliquity and precession bands. This indicates that the nonlinear response of the system is effectively amplified with this new parameterization of albedo. In addition, the comparison of the time series (Fig. 16a and g) shows great correlation, indicating that both the amplitude and timing of the GIV are well captured. The evolution of $V_{ice}$ matches the proxy record qualitatively well despite its simple representation.

In this model configuration, deglaciations start because the first term of Eq. (27) (i.e. the effective shortwave radiation $I_{\text{eff}}$, Eq. A1) increases ablation thanks to the aging in ice albedo (Fig. A2). At a certain time this contribution is high enough to outweigh snowfall, and the glacial termination starts. This process is enhanced with the ice-sheet surface elevation at the end of the glacial termination. This phenomena can be seen in Fig. 17, where we have represented the trajectories of $H$ as a function of the insolation forcing. This diagram explicitly shows the nonlinearities in the system: ice starts to grow when insolation becomes low enough to allow its persistence over several precession and obliquity cycles, until the effective shortwave $I_{\text{eff}}$ outweighs the accumulation. Then, a glacial termination is triggered.

## 3 Discussion

What is the minimum number of required physical processes necessary for satisfactorily simulating glacial-interglacial cycles? By sequentially increasing the complexity of our simple model, we have evaluated the mechanisms that ultimately facilitate capturing the right timing and amplitude of glacial cycles (Table 4).

We have seen that the last configuration of PACCO (AGING experiment, Sect. 2.7) produces results that highly correlate with various paleoclimatic records. The mechanisms that trigger GIV in this configuration are in agreement with state-of-the-art hypotheses (Ganopolski and Calov, 2011; Abe-Ouchi et al., 2013, among others). The 100 kyr periodicity is produced due to both ice sheet dynamics and interactions with the climate. On one hand, ice-sheet velocity determines the responsiveness of the ice sheet to the insolation forcing. The change in $C_s$ affects the level of ice sheet sliding and thus facilitates its synchronization with the forcing via ice discharge. Therefore, the periodicity of the system response varies with $C_s$. On the other hand, isostasy and ice aging provide two essential feedbacks that trigger lagged deglaciations. Both processes become more and more important when the ice persists in time. When the albedo is low enough, an increase in insolation can trigger a deglaciation that is reinforced with the ice surface elevation feedback. We have also seen that ice-sheet thermodynamics could play an important role in this variability since it increases the ice basal velocity. However, in our case ice aging is more effective, because it introduces a mechanism that persists over multiple glacial cycles.

Another advantage of PACCO is that it is not necessary to perform a strict calibration of each model parameter to obtain GIV. Of course, we could have performed thousands of permutations to obtain the perfect run. But that is outside the scope



**Figure 16.** Results of AGING simulation for $\tau_\alpha = 1000$ yr. Note that the initial conditions are applied at 900 kyr BP only to let the model equilibrate.

of this paper, since our purpose was to build a physically explicit conceptual model with which to study the dynamics of the climate-ice sheet interaction.

Finally, the sensitivity of the ice sheet to sliding remains to be explained and attributed. One possible mechanism could be the rigidity of the substratum on which the ice sheet is formed, as proposed by the regolith removal hypothesis (Clark and Pollard, 1998; Ganopolski and Calov, 2011; Ganopolski, 2024) in the context of the Mid-Pleistocene Transition. This mechanism, together with the full Pleistocene problem, is reserved for future work.





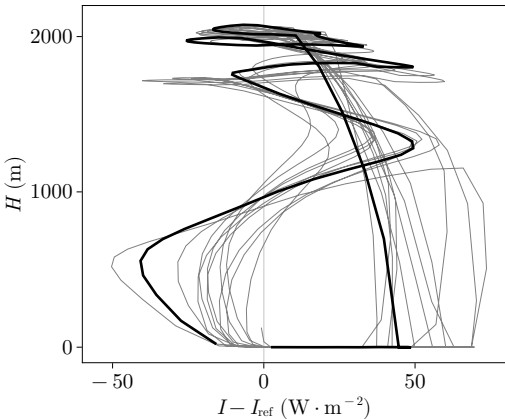

**Figure 17.** Trajectories of the AGING experiment, we highlight the last glacial cycle in black.

**Table 4.** Summary and main conclusions of the experiments performed in this work.

| Experiment | Conclusion |
| --- | --- |
| LIN | Linearly forced mass balance system behaves nonlinearly due to ice sheet dynamics |
| NONLIN | The system reacts at periodicities greater than the insolation forcing for moderate and weak basal ice-sheet dynamics |
| ISOS | Isostatic delayed rebound slows down the mass balance response of the system and facilitates terminations regardless of the intensity of basal sliding |
| RISOS | By forcing the system with the real insolation, resonance at greater periodicities than the forcing appears, but the very simple climate formulation does not allow for a satisfactory calibration with the real timing of the cycles |
| BASE | Introducing am improved ablation scheme, the effect of the ice sheet size on the regional temperature and a simplified carbon cycle allows better agreement with proxies |
| THERM | Including ice thermodynamics and its effects on basal sliding favors the emergence of 100 kyr periodicity but still shows a poor agreement with the terminations timing shown by the proxies |
| AGING | Ice darkening enhances ablation slowly and finally provides a very good simulated timing and amplitude of glacial cycles |

## 4 Conclusions

Here we have developed a simple physically-based model in order to sequentially identify the mechanisms responsible for GIV
of the last 800 kyr. Our model is novel in the way it is formulated because the equations related to ice dynamics are obtained
through spatial lumping of ice-sheet modeling equations. We have seen that in PACCO features of Late Pleistocene proxy
records are reproduced due to both ice-sheet dynamics and interactions with the climate. The delayed isostatic response adds a
slow component to the system that generates asymmetry in the cycles (in agreement with Abe-Ouchi et al., 2013). Moreover,
the aging of ice due to its natural darkening (given by compaction and dust deposition) provides an additional mechanism



to the ablation rate, which allows glacial terminations when the ice sheets are big enough to cool the climate. This trigger is activated when the insolation reaches a certain threshold in which melting outweighs snowfall. The combined contribution of isostatic rebound and ice darkening explain the 100-kyr paradox and allow reproducing the power spectrum of glacial cycles along with good timing for glacial terminations with respect to the paleo climatic records (in agreement with Ganopolski and Calov, 2011).

All these conclusions are subjected to the simplicity of the model, regarding the accuracy of the results and the taken approximations. However, to our knowledge, PACCO is the first conceptual model to explicitly resolve the most important processes and to produce the Quaternary glacial-interglacial cycles with minimal physics. It opens a new way of conceptual modeling that allows different hypotheses to coexist and be isolated from one another.

*Code and data availability.*  PACCO is available at https://github.com/sperezmont/Pacco.jl. The archived version of the code in this paper
can be found at https://doi.org/10.5281/zenodo.11638682. The code to generate all the figures of the document and its archived version can be found at: https://github.com/sperezmont/Perez-Montero-etal_YYYY_ESD and https://doi.org/10.5281/zenodo.11639519.

## Appendix A: Further analysis of the AGING experiment

### A1    AGING sensitivity to sliding factor $C_s$

A sensitivity experiment with respect to the sliding factor $C_s$ was performed for AGING configuration (Fig. 14). Fig. A1c
shows that a value from $10^{-10}$ to $10^{-7}$ (Pollard and DeConto, 2012) produces asymmetric oscillations similar to those that would be expected from GIV.

### A2    Mechanisms of glacial termination with the AGING configuration.

The effective insolation received by ablation is defined from Eq. (27) as

$$I_{\mathrm{eff}} = (1-\alpha) \cdot (I - I_{\mathrm{ref}}), \tag{A1}$$

which is limited to positive values. $I_{\mathrm{eff}}$ is modulated by albedo aging (Fig. A2b), therefore, the contribution of shortwave radiation to mass balance increases with time. In this way, if the ice persists enough time, shortwave contribution can outweigh snowfall and thus initiate a glacial termination.

*Author contributions.*  JAS and SPM conceived PACCO. SPM implemented PACCO, performed the analysis, created the figures and tables, and wrote the paper. JSJ improved the code efficiency and structure. DMP largely contributed to conceptualise the governing equations of
ice-sheet thermodynamics. JAS, JSJ, DMP, MM, and AR provided extensive feedback on the analysis and the article.



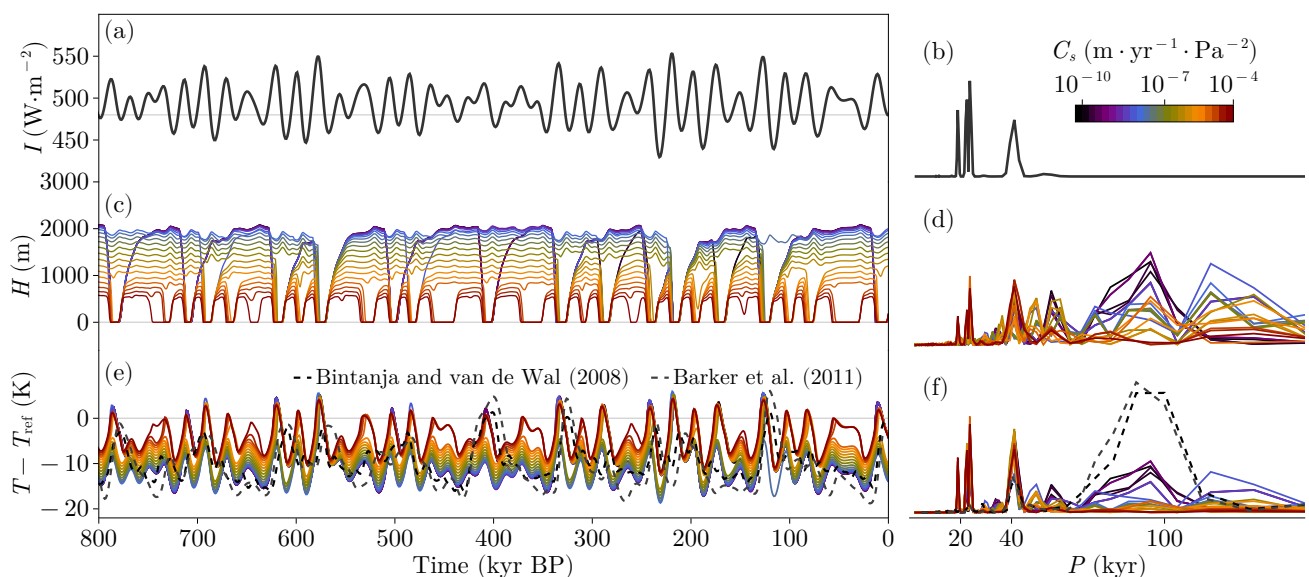

**Figure A1.** Results of an AGING experiment using different $Cs$ values. (a, c, e) Time series obtained from the model using different sliding factors. (b, d, f) Periodograms obtained from the time series in the left column. Note that when normalizing spectra, series were cut off for periods larger than 200 kyr.

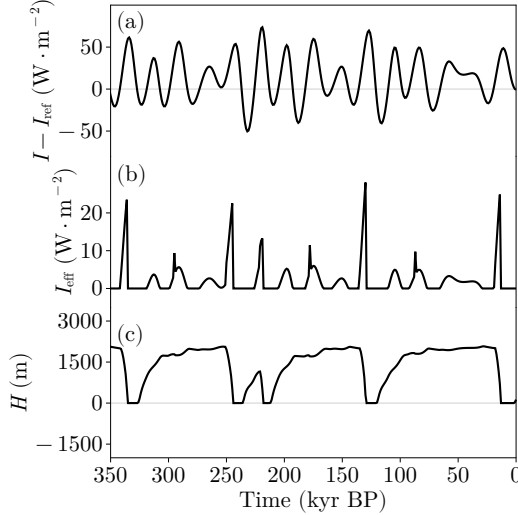

**Figure A2.** (a) Evolution of insolation, (b) effective shortwave radiation and (c) ice thickness in the AGING experiment.





*Competing interests.* The authors declare no competing interests.

*Acknowledgements.* We thank Lucía Gutiérrez-González, Félix Garcia-Pereira and Nagore Meabe-Yanguas for their helpful suggestions on the design of some figures.

*Financial support.* This research has been supported by the Spanish Ministry of Science and Innovation (project IceAge, grant no. PID2019-110714RA-100 and project CCRYTICAS, grant no. PID2022-142800OB-I00). AR received funding from the European Union (ERC, FOR-CLIMA, 101044247).




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
