# Peer review of "A simple physical model for glacial cycles"

_EGUsphere, 2024_

## Community Comment (CC3)

Dear Sergio Pérez-Montero, Dear Colleagues,

Thank you for considering my comments.

1. Unfortunately, your equation CC1.1 is also problematic because it implies that ice sheets that do not have a boundary with an ocean ($L_{ocn} = 0$) have ice discharge $q \to \infty$.

Let us return to the basics. The total mass balance of an ice sheet can be written as

$$\frac{\partial(HS)}{\partial t} = \dot{m}S - \oint_\gamma \int_0^{h_g} v_g \, dz \, d\gamma$$

Here $S \sim L^2$ is the area of an ice sheet, $\gamma$ is the grounding line, $h_g, v_g$ are ice thickness and normal velocity at the grounding line, correspondingly. Indeed, the parameterization of the term $\oint_\gamma \int_0^{h_g} v_g \, dz \, d\gamma$ is a difficult task because at the grounding line ice thickness $h \to 0$, but ice slope $\partial h / \partial x \to \infty$. You are not the first ones who face this challenge. For example, in 1980, as a PhD student, I had to present my work to G. I. Barenblatt, and I was honored to be schooled by him about this term. I am not going to advocate here necessarily for the parameterization that was adopted as the result of this conversation (we can always discuss it off-line if needed), but if you want to continue to scale the above mass balance the same way as you did before, then your mass discharge $q$ should be proportional to

$$q \sim \frac{L_{ocn}}{L^2} vH$$

2. I am afraid my advocacy of the vertical temperature advection has been misunderstood. Within the basal boundary layer the vertical temperature advection is indeed negligible. What I meant is that regional air temperature $T$, that goes into calculations of your $h_{cond}$ is not instant $T(t)$ but it is temperature "from the past", i.e., $T = T(t - \tau)$ where $\tau = H/\dot{s}$ is the vertical-temperature-advection timescale, i.e. time of the temperature "delivery" from the ice-sheet free surface to the upper boundary of the basal boundary layer.

Respectfully,

Mikhail Verbitsky

---

## Author Response (AR1)

**Summary of changes made to the manuscript after the first round of revision**

In this document we summarize the changes made on the manuscript as a response to the reviewers and the community comments. For specific details on the discussion, we refer the reader to our responses: https://egusphere.copernicus.org/preprints/2024/egusphere-2024-1842/#discussion (AC1-AC5)

**a. Notes on minor changes**

The authors found and corrected some errata when revising the new version of the article.

**b. Changes made section by section**

**Abstract**

Nothing to note.

1. **Introduction**

   - The attribution of the astronomical theory of glacial cycles has been reformulated.
   - Some new references are added.
   - The explanation of the conceptual approach to solve the 100-kyr paradox has been rephrased.
   - It is now clearly specified in what sense we mean that our model is adimensional and its purpose.

2. **Model description and results**

Overall, a detailed description of the physical meaning of each equation and its justification is provided, as described in AC5. Now we will describe specific changes (note that we refer to the equations in the new manuscript):

   - Equation 2: Ice discharge equation now depends on parameter $L_{ocn}$ (AC1).
   - Equation 7: Driving stress is now dependent on a dynamical L (AC1). And L is defined in Eq. (8).
   - Equation 11: Now we make a distinction between basal and deformational regime, thus the basal stress depends on parameter $\theta$ that represents the slope of the streaming region.
   - Table 1: For ease of reading, a more detailed description of each experiment is presented.
   - Model configuration schemes (Figs. 2, 4, 6, 10, 12 and 14) present a description of the specific equation they use.

   2.1. **A quasi linear configuration for surface mass balance (LIN experiment)**

   - Equation 13: We have eliminated the contribution of eccentricity for simplicity.
   - Table 2: Now, all model parameters are gathered in the same table and sorted in alphabetical order.

   2.2. **Introducing feedbacks on surface mass balance (NONLIN experiment)**

Nothing to note.

**2.3. Delayed isostatic adjustment (ISOS experiment)**

Nothing to note.

**2.4. ISOS configuration with real insolation (RISOS experiment)**

Nothing to note.

**2.5. Improving the coupling between ice sheet and climate (BASE experiment)**

Nothing to note.

**2.6. Ice-sheet thermodynamics (THERM experiment)**

- The equations of this section discussed during AC1 and AC2 have been properly modified and justified.
- Since now THERM configuration depends on less parameters, we only show the sensitivity experiment for $\tau_{kin}$ in this section. The sensitivity test for $C_s$ is shown in the new Appendix A1.
- Figure 13 has been changed according to the previous point.
- The partial conclusions on this section have been modulated and nuanced according to the community comments of M. Verbitsky (AC1-AC3) and the comments of reviewer M. Crucifix (AC4).

**2.7. Ice aging (AGING experiment)**

- Equation 40: A personal communication from Andrey Ganopolski brought to our attention that a simplified version of this equation can be made. Now, the albedo evolves as a diagnostic equation and explicitly shows the characteristic time scale involved in it.
- We have also changed the design of Fig. 15 to establish an analogy with Fig. 13 and simplify the message.
- Equations 43 and 44: Since we introduced a specific ice profile (AC1-AC4) using a typical ice sheet scale L dependent on ice thickness H, the diagnostic equation for the ice surface needed modification. Thus we concluded that equations 43 and 44 qualitatively represent the processes that allow an ice sheet to be more or less extensive.

**3. Discussion**

As suggested by the reviewers, we extended the discussion and now we guide it point by point based on the insight gained by adding different levels of complexity to the model.

**4. Conclusions**

We reformulated some conclusions as suggested by the reviewers in order to be more specific with the model capacities.

**Appendix A: Further analysis of the THERM and AGING experiments**

**A1. THERM sensitivity to sliding factor $C_s$**

As pointed out above, we included this section to provide a sensitivity experiment of THERM configuration to $C_s$.

**A2. AGING sensitivity to sliding factor $C_s$**

Nothing to note.

**A3. Mechanisms of glacial termination with the AGING configuration**

Nothing to note.

**c. Point-by-point response to reviewers' and community comments**

Attached below are the responses to the reviewers and the community.

**Response to Michel Crucifix referee comment (RC1)**

Dear Michel Crucifix,

Thanks for your comments. We hereafter respond point by point:

*Review of « a simple physical model for glacial cycles » by Sergio Pérez-Montero.*

*1. General impressions*

*The contribution of the authors is a very welcome addition to the literature of models of glacial-interglacial cycles. The model that they introduce, called PACCO, features 0-d ice sheet dynamics along with a simple linear predictor of $CO_2$ concentration. It is argued that it is as an adequate device for "testing the different hypotheses and isolating them from each other".*

*1.1 Mechanism attribution*

*How much we learn about physics and balance of mechanisms in a simple model is an interesting question. I would suggest the following argument. Consider a model with thousands of lines of codes, with the representation of highly detailed mechanisms based on physical principles and parameterisations. Suppose then that this model successfully reproduces one or several glacial cycles. For sure, such achievement required some tuning (the Saltzman 1990 argument), but one might argue that, in such a big model, the relative importance of different components or mechanisms in the resulting ice age dynamics is fairly robust. It does not mean that it is faithful to the truth, but that it is hard to change this balance by tuning. Now consider a low-order model with, say, 10 state variables such as PACCO. One might suspect that different choices of parameterisations or even different assumptions may have a large impact on the relative role of different components. This is both the strength and the weakness of the low-order models. A strength, because it offers opportunities of experimentation and addresses sensitivities not apparent in the bigger model; a weakness, because as different assumptions might converge to similar mathematical expressions, interpretation is not as straightforward as it may look. This modelling ambiguity was I would argue perhaps most severe in Saltzman - Maash models (e.g. Saltzman and Maash 1990, hereafter SM90), where only one out of the three differential equations was non-linear (the one for the carbon cycle); the carbon cycle was therefore by design given the responsibility of causing the limit cycle.*

*For sure, PACCO has more physically explicit mechanisms than SM90, but I suspect caution must still be exercised when attributing a "role" to a component --- I will make a case below that perhaps the respective roles of isostasy, basal melt and snow/ice aging might be sensitive to questionable assumptions (through comments 1.2 and 1.3).*

*Before going further, I would suggest the authors to have a look and perhaps position themselves with respect to two articles which they did not cite, yet I suspect that have been somehow influential --- because some of the comments I am making here have already been made there, and also because of snow aging was already the attention of a controversy at the time:*

*- Gallée et al 1992 introduced what they called a "sectorially" averaged climate model, with different parameterisations affecting the heat balance of the ice sheet including snow aging (based on previous works by L.D.D Harvey --- also relevant for your section 2.7)*

*- Tarasov and Peltier 1997 reported a simulation of the latest termination with an energy balance model coupled to an ice sheet model. They commented the Gallée et al. 1992 paper as well as other, similar efforts, and specifically, they made the following point: "Although the addition of each of these plausible mechanisms [they quoted here the list of mechanisms] to the reduced model has led to improvements in their*

*ability to acceptably simulate the ice age cycle, it has remained unclear as to whether difficulties encountered may not be due to an incorrect representation of the processes already included [.].*

*It could be unfair to take this quote out of context, but I believe the Tarasov and Pelter question is legitimate: if a model includes a number of processes (in PACCO: we have ice sheet dynamics, lithosphere adjustment, snow ageing, parameterization of ice discharge itself obtained depending on a parameterization of the bottom balance), each of these processes being necessarily highly idealised, what robust conclusions can be obtained about the relative importance of each of these processes in the emerging phenomenon which is, in this case, the amplitude, timing and shape of glacial interglacial cycles?*

*Let us be fair: the authors of PACCO do not go as far as quantifying the relative importance of each mechanism. However, if I read well, they attribute the asymmetry of cycles to the delayed isostatic response, and give a central role to ice aging for the timing of glacial-interglacial cycles.*

*Verbitsky et al. 2018 (VCV18) includes many fewer processes of PACCO and so cannot possibly investigate the relative importance of these different mechanisms (as the authors of the present contribution correctly pointed out). For example, it ignores isostasy and snow ageing. For sure, not representing a mechanism does not imply that it plays no role. But nevertheless the question remains: how is it that VCV18 produces glacial interglacial cycles with decent amplitude, timing and shape with only ice flow dynamics, basal temperature feedback and a general, linear climate feedback, while PACCO needs isostasy and snow ageing ? One might argue, VCV18 implicitly represents the isostasy effect in its equations. VCV18 authors were aware of this, and this is the reason why they pitched their paper by emphasizing the role of adimensional numbers, specifically with their attempt to define a "V-number" measuring the ratio between positive and negative feedbacks.*

We understand your argument and we mostly agree. In a simplified model like PACCO, isolating the relevant processes and illustrating their relative importance is subjected to the potential lack of accuracy inherent to reducing the problem to a few differential equations without explicitly solving the spatial dimensions. To illustrate this, one could think of a deglaciation of the Northern Hemisphere ice sheets: it is likely that the ice darkening process would be highly relevant for the southern ablation borders of the ice sheets, whereas the effects of a basal thermally driven acceleration of ice streams would be more relevant for inner parts of the ice sheets, both of them finally contributing to the nonlinearities at the core of deglaciations. This spatially heterogeneous phenomenon would not be captured by a model with reduced spatial dimensions as PACCO, in which solely the "mean" state of the ice sheet is simulated. Therefore we agree that conclusions on the relative importance of the mechanisms have to be drawn with caution and we have expanded on this in the new version of the manuscript.

That being said, we have built PACCO by including all the suspected relevant processes for glacial cycles in the simplest, yet keeping a physically-based approach. By doing so, we believe that we have illustrated in a novel way how progressively including different mechanisms translates into the gradual emergence of periodicity peaks of the climate-cryosphere response that are not present in the insolation forcing alone. However, we were perhaps not explicit enough on how well VCV18 already captured the timing and shape of glacial cycles and we have better acknowledged this in the new version of the manuscript. Indeed, PACCO's THERM experiments also perform satisfactorily (see former figure 13), and have now been considerably improved thanks to the implementation of the suggestions made by Mikhail Verbitsky (see figure CC2.1 in our response to CC1 and CC2 comments). Nonetheless, as described in the manuscript, the inclusion of a progressive decrease in surface albedo resulting from ice aging facilitates in our model a good calibration of the glacial cycles rhythmicity shown in proxies.

Following your suggestion on mechanism attribution, we have now added the references to Gallée et al, 1992 and Tarasov and Peltier 1997.

**1.2 Mechanism for the delayed feedback**

*When it comes to the "asymmetry" of cycles, I suspect the crucial element is the presence of a delayed feedback (see, e.g., Quinn et al. 2018 who coded a glacial-interglacial model with delay equations). What component generates delayed feedback ? True, isostasy does. But the thermal balance of the basal ice does, too. On this point, there is one concern that has already been mentioned in the public discussion by M. Verbitsky. Equation 34 linking the conduction heat flux to the surface air temperature is disputable. The point of controversy here relates to propagation time needed by air temperature signal to reach the bottom layer, which involves an advection timescale. This effectively introduces a delayed feedback which can be crucial for the generation of a high amplitude glacial cycle with it its asymmetrical shape.*

We agree. Indeed, in our simulations the elevation feedback (NONLIN set of experiments) as well as isostasy (ISOS) favor the emergence of the asymmetry, which is considerably enhanced when considering the effects of the thermal basal state on ice sliding (THERM) by means of a delayed feedback. The characteristic time for the base to become temperate, thus enhancing sliding, depends on the vertical diffusion within the ice column, the geothermal heat flow and the vertical advection. This is better captured with the improved version of THERM thanks to M. Verbitsky's suggestions (see details in CC2). Using this approach the base gets tempered relatively fast as the analytical study of Moreno-Parada et al. (2024) shows. The delayed feedback associated with the basal thermal state is now largely explored in terms of the (dynamic and dependant on accumulation rate) Péclet number and the relaxation time of ice-stream activation when the base becomes temperate. Thus, even though the thermodynamics is now more robust our conclusions remain similar. The new version of the manuscript will reflect these improvements. Please see also the responses to comments CC1 and CC2 and also figure CC2.1

**1.3 Fixed-length assumption**

*Similar, the fixed ice sheet length (L) assumption may significantly distort the albedo feedback and all feedbacks associated to elevation (I think this assumption would affect all dynamical aspects of the model) I'd like to confirm here that I identified these concerns before actually reading Mikhail Verbitsky's comments, though it is not surprising that we both spotted this issue given our common modelling history, and, to be clear, I haven't inspected the discussion that followed the original community comment.*

As you say, fixing L in the ice-sheet dynamic equations overestimates the driving stress during the second half of the cycle and thus changes ice velocity. Thus, as M. Verbitsky suggested, we introduced an ice profile of the form $L = c * H^2$. We have therefore changed the formulation of deformational velocities accordingly. Please see AC1, AC2 and AC3 for more details. The new version of the manuscript includes these corrections.

*Comments 1.1, 1.2 and 1.3 do not by any means question the legitimacy of the present study but they come as a word of caution about what one should view as the most adequate device for "testing the different hypotheses and isolating them from each other". I would argue that for that aim the device must have the output and resolution needed to be confronted with detailed observations which are independent enough, thus providing independent constraints. Only in this case, I would argue, is mechanism attribution less ambiguous. Besides these concerns (heat flow parameterisation, fixed length, and caveat about mechanism attribution), most comments below can easily be addressed because they are of editorial nature.*

We acknowledge all your comments. We have now modulated and nuanced our discussion and conclusions with respect to attribution to specific mechanisms when preparing the new version of the manuscript.

*1. Introduction*

> *- Some comments about the introduction have already been made by Mikhail Verbitsky. Just adding here that Milankovitch did not "postulate" that glacial - interglacial variability is caused by changes in the insolation. Milankovitch can be seen, as summed up by Berger (2021), as the father of (paleo-)climate modelling. He reasoned on the physics of the radiative balance and effectively computed the variations in temperature to obtain an "effective snow line" (expressed as a latitude) caused by the quasi-periodic changes in insolation. The "postulate" about the astronomical origin of ice ages has to be found earlier (specifically, speaking of the attribution of ice ages to summer insolation , J.J. Murphy 1876. See reviews by Berger 1988 and Berger 2021.)*

Thank you, we have corrected this in the new version of the paper.

> *- The phrase "conceptual model" is widely used, but it may be a bit ambiguous. Perhaps use the distinction between "phenomenological" (describing the observations in a way consistent with the theory but not derived from it --- Paillard 1998 would be phenomenological, for example) and "theoretical" models? In that sense PACCO is indeed a theoretical model with some empirical parameterisations; Verbitsky et al. 2018 also present their model as a theoretical construction. There is no sharp line between phenomenological and theoretical, but I suspect this distinction would help for the presentation of PACCO.*

Thank you for the suggestion, we have made the distinction between phenomenological and theoretical where appropriate.

> *2a. Model description*

> *- Would it be possible to have all parameters listed in one table by alphabetical order? It was at places a bit painful to find the definition and value of the different parameters while reading the text (e.g., after equation 16, or to that \dot s and k_{\dot s} are model parameters, but doesn't mention that T_ref is also listed in the parameter table. It took me a while to find T_thr which indeed is also in table 2.*

We acknowledge your recommendation and we will remake all tables to include all parameters in alphabetical order in just one table in the next version of the manuscript.

- Synthetic insolation forcing: I found introducing P_e and then setting it to zero confusing. Why not simply explain that one major difference between the synthetic signal and the real one is the amplitude modulation of precession by eccentricity? (Which indeed can be relevant for resonance phenomena)

We agree and have followed your suggestion when preparing the new version of the paper. We initially included it just to show the complete approach and the possibilities of the model, but as you say, it is cleaner and more elegant to eliminate that contribution for the sake of simplicity and clarity.

> *- line 300 : I did not understand what is meant by the "Dynamic nature of thermodynamic hypothesis".*

We intended to say that the thermodynamic hypothesis (i.e. the base becomes temperate and then basal sliding is increased) affects the ice-sheet dynamics in the sense that if you try to produce glacial cycles via this hypothesis, you need a system that reacts highly to changes in the base. If the system is not sensitive enough to these changes, then glacial cycles are not produced in a realistic manner, at least with our formulation. We will rephrase this to make it more clear.

*2b. Result analysis*

*- line 216 : replace "higher periodicity"  by longer periods.*

Done.

*I suspect (but this can be discussed) that the word "resonance" is used a bit abusively here. Resonance would refer to a disproportionate increase in amplitude for certain frequencies of the forcing (and then distinguishing linear from non-linear resonance, non-linear resonance occurring when the matching frequency depends on the amplitude of forcing). It is not phase locking either because the model does not have a self-sustained and oscillation. So I believe the right phrase here is just non-linear response with the output period being a multiple of that of the forcing.*

Agreed.We will change that phrase following your suggestion.

*- line 220: "on the contrary, even a moderate increase in insolation induces a termination too easily". (If this is indeed what you meant; though it might be argued, high sensitivity to insolation is also what you need for the MIS12-11 transition).*

We agree, you need a high sensitivity to insolation to produce MIS11, however, in this particular model configuration (ISOS and RISOS), a high sensitivity is maintained all along the simulation. Thus, the energy in 20 and 40 kyr is very high and PACCO does not produce 100-kyr glacial cycles. It is only after including a more complex climate representation (after the BASE configuration) that sensitivity to insolation evolves dynamically in time so that the model reproduces MIS11.

*- Line 370: the sentence is a little bit unclear. Isn't it normal to have high sensitivity to sliding?*

*In the sentence "one possible mechanism could be related to the rigidity of the substratum on which ice is formed" : please clarify or rephrase: rigidity is not a mechanism.*

We understand that "rigidity" was not a good choice to refer to the regolith hypothesis (Clark and Pollard, 1998) and we will be more explicit in the new version of the manuscript.

*- Figure 13: For clarity just specify which h_geo was used for the I,k,m,o panels.*

We will specify that in the new version of the manuscript.

Sincerely,

Sergio Pérez-Montero et al.

*References:*

*Gallée H., J. P. Ypersele, T. Fichefet, I. Marsiat, C. Tricot and A. Berger (1992), Simulation of the last glacial cycle by a coupled, sectorially averaged climate-ice sheet model. Part II : Response to insolation and CO2 variation, Journal of Geophysical Research, (97) 15713-15740 doi:10.1029/92JD01256*

*Saltzman B. (1990), Three basic problems of paleoclimatic modeling: a personal perspective and review, Climate Dynamics, (5) 67-78 doi:10.1007/BF00207422*

*Saltzman B. and K. A. Maasch (1990), A first-order global model of late Cenozoic climate, Transactions of the Royal Society of Edinburgh Earth Sciences, (81) 315-325 doi:10.1017/S0263593300020824*

*Tarasov L. and R. W. Peltier (1997), Terminating the 100 kyr ice age cycle, Journal of Geophysical Research, (102) 21665-21693, doi:10.1029/97JD01766*

*Berger A. (1988), Milankovitch theory and climate, Reviews of Geophysics, (26) 624-657*

*Berger A. (2021), Milankovitch, the father of paleoclimate modeling, Climate of the Past, (17) 1727-1733 doi:10.5194/cp-17-1727-2021*

*Quinn C., J. Sieber, A. S. Heydt and T. M. Lenton (2018), The Mid-Pleistocene Transition induced by delayed feedback and bistability, Dynamics and Statistics of the Climate System, (None) None doi:10.1093/climsys/dzy005*

References:

Clark, P. U., & Pollard, D. (1998). Origin of the middle Pleistocene transition by ice sheet erosion of regolith. *Paleoceanography*, *13*(1), 1-9.

Moreno-Parada, D., Robinson, A., Montoya, M., & Alvarez-Solas, J. (2024). Analytical solutions for the advective–diffusive ice column in the presence of strain heating. *The Cryosphere*, *18*(9), 4215-4232.

**Response to Anonymous referee comment ([RC2](RC2))**

Dear Anonymous referee,

Thank you very much for your comments. We hereafter respond point by point:

*Authors investigated Pleistocene glacial cycles that are known to be driven by the nonlinear response of the climate to solar forcing. Though complex models accurately simulate these cycles they are computationally expensive. Simpler conceptual models lack physical detail, though. The Physical Adimensional Climate Cryosphere mOdel (PACCO) used in this study aims to balance complexity and simplicity by focusing on the interaction between climate and Northern Hemisphere ice sheets. Authors show that PACCO effectively reproduces 100,000-year glacial cycles by incorporating ice-sheet dynamics, thermodynamics, and ice aging. The study reveals that ice aging and delayed isostatic response are key to matching geological records, making PACCO a valuable tool for studying glacial cycles.*

*Major comments on the paper:*

*I have to say that I like how the paper was written. It goes from a simple model, where linear insolation forcing is introduced to the one-dimensional model as described by SIA equations. I do not agree with RC1/CC1's description that it even might have 2 dimensions, as here H is not a dimension but merely a product of equations. I only miss here the motivation for all constants and order of complexity they introduced to the model. For example, in table 2, Length is 1000 km based on Oerlemans et al., 2008. But why is simply that taken and not given a reasoning? The same goes for many equations that are not fully motivated. Please note that I am a big fan of simple conceptual models as I believe that we can learn more from simple yet fast models compared to expensive and long-time running Stokes models.*

We thank you for your comment. The motivation behind the gradual increase in complexity is to obtain glacial cycles with good amplitude and timing with respect to the proxies. For that, we analyzed each step in order to fully understand in which way each process affects the variability of the system. We understand that this intention might not be as clear as we thought and we will elaborate more on the justifications in order to make it more clear. For that, we have extended the justification of the following equations:

- Equation (1) is a mass conservation equation that takes into account mass inputs and outputs, similar to the one employed in Benn et al. (2019).
- Equations (2) and (3) are combined into one equation with explicit definition of each component. We also added a new equation (based on the comments from CC1, CC2, CC3 and RC1) that explains the typical horizontal scale of an ice sheet and the approach followed to obtain the parameters employed.
- Equations (7) and (8) are now related with the basal stress in a different manner. In the new version of the manuscript we have an equation for the driving stress and another one for the basal stress. The reason is basically that the ice profile associated with those dynamic regimes is different and two equations are needed if we are to follow the approach suggested by M. Verbitsky in CC1. After presenting the equation we explain that the ice profile of a region dominated by basal dynamics is mostly horizontal and, thus, a constant slope in the basal stress is assumed.
- Equations (9) and (10) are justified now explaining that they represent how the ice deforms under its own weight.
- Equation (16) is motivated by the effect of temperature on the amount of allowed moisture in the atmosphere: more temperature, more moisture.

- Equation (21) is now justified by explaining the interaction between the ice and the bedrock. If the ice sheet grows, the bedrock sinks. However, this is not instantaneous but takes a certain amount of time given by the relaxation time, which is a parameter of the model.
- Equations (29-34) have been simplified and explained in detail:
  - We introduced the notion of Péclet number, its relation with Hb and H as suggested by CC2.
  - We have included the advective heat contribution and explained its formulation.
  - We have added a phrase that summarizes the purpose of the entire THERM configuration, that is basically "to mimic the warming at the base of the ice sheet and to slow down its effect in order to represent how this mechanism is propagated through the entire ice sheet."
- Equation (38) and (39) have been simplified to one diagnostic equation that clarifies how albedo ages in time. This change was made thanks to a personal communication from A. Ganopolski.

*Then, from section 2.2 onwards, authors speak of periodicities at time scales of 60-80-100-120 kyears. I must disagree as, e.g., Figure 5 shows pronounced peeks around 20 and 40 kyears for all values of the sliding parameter, and around 60 kyears for lower values of the sliding parameter. Above that, there is no significant peak which might suggest that what we see is just a model artifact or an aliasing error. And this is additionally supported with the figure 5c. The same holds for figure 7 (ISOS simulation). When we go to figure 9 (RISOS simulation), the authors statement does not hold at all, because the peaks really depend on the sliding parameter and cannot be generalized. So here, for me, the result of AGING simulation is the only result that can claim that there is a periodicity between 80-120 kyears (the correct time really depends on the sliding parameter).*

We agree in the sense that it is true that the more remarkable peaks are found at 20 and 40 kyr. However, when we mention multiples of those periodicities we refer to the appearance of peaks around those multiples, but not its dominance over 20 and 40 kyr. In this way, we tried to show how adding different mechanisms favors the appearance of longer periods. This can be seen when inspecting the results of the different experiments in order of increasing complexity. We understand that our intention was not as clear as we thought and we will clarify this issue in the new version of the paper.

*Another thing that should be changed in the following manuscript is explanations and captions of all figures. For example: we can see a change in dominant peaks which is logical as the authors introduce the model complexity. However, it is not explained why the energy of the peaks for the same sliding factor changes between the runs (see for example the difference in peaks at 60 kyears between Figure 5 and Figure 7. Also, here the vertical axes would really be beneficial to make a good comparison.*

We did not add them in the first place since we think the comparison must be made within each configuration and between different values of the sliding parameter, as the model changes between configurations. Nonetheless we include a figure below with the absolute PSD (power spectral density) of every experiment (Fig. 1). The comparison between configurations should only be made in the appearance or not of periodicities longer than the ones provided by the forcing, always having in mind the evolution of the variables in time. This comparison allows us to illustrate how, as you pointed out above, adding complexity increases the nonlinear response, which is basically the aim of the paper structure.

[Figure]

**Figure 1.** Power spectral density of ice thickness ($H_{PSD}$) for each experiment performed. Note that the vertical axes are different for each panel and that the colors change according to each experiment. Note also that these are the results after the comments following the open discussion and both referees.

*Additionally, the authors did explain why we see differences in periodicity between the sliding parameters, and why their statements are "true" for the lower values of the sliding parameter. There to say, for which glacier could we use these results to possibly predict the future evolution?*

Our model was conceived to reproduce the interaction between Northern Hemisphere ice sheets and climate. Different modeling studies (e.g. Clark and Pollard, 1998; Willeit et al., 2019; among others) rely on the regolith hypothesis. This hypothesis basically indicates that a gradual reduction in the capability of the ice sheets to slide during the Pleistocene made them slow and less dynamic, leading to larger ice sheets and a nonlinear impact on late Pleistocene glacial cycle frequency. In our study we find this to be true for AGING configuration. When the sliding capacity is not very high, our model is capable of synchronizing more with the paleoclimatic proxies. To answer your last question, PACCO could simulate the evolution of the Northern Hemisphere in the long-term future.

*Thus, to conclude: I think the paper is a valuable contribution to understand the glacier cycles and ice-sheet dynamics. However, I would not consider it for publication just yet. 1. The model and its equations need to be explained and introduced better as as CC1 says: "it looks like a verbalized computer code". 2. The figures are not "self-standing" and need a better explanation of all lines, better comparison between experiments, and y-axes would be beneficial. 3. Better discussion is necessary as it is not obvious how this model can be used and what is a future of it.*

Thank you for your final comments which are very useful to better formulate the message of the manuscript. To this end we will:

1. Explain and justify more deeply every equation in the paper.
2. Describe better the figures and add the vertical axes to periodograms to allow a better comparison between experiments.
3. Add a deeper explanation of the possible applications of the model such as the research about the Middle Pleistocene Transition or the Anthropogenic future.

Sincerely,

Sergio Pérez-Montero et al.

**Response to Mikhail Verbitsky comments (CC1 CC2)**

Dear Mikhail Verbitsky,

Thanks for your comment. We hereafter respond point by point:

**CC1**

*Simple but physics-based models of ice ages have been introduced many years ago (e.g., Birchfield and Weertman, 1978; Chalikov and Verbitsky, 1984, etc.); nevertheless, phenomenological models of glacial rhythmicity are, as Michel Crucifix (2012) said, still "seductive". Their obvious lack of underlying physics is compensated by introduction of artificial thresholds and other Boolean statements that, indeed, allow them to reproduce empirical data but do not add much to our understanding of the physical nature of glacial periods. Therefore, every attempt to introduce "A simple physical model for glacial cycles" should be welcomed.*

*Unfortunately, the presented paper has (I frankly hesitate to say that, but for the lack of a better word…) a flaw that needs to be addressed before one proceeds to the results.*

*You approximate ice discharge as q = v H/L, where L is a constant. Even intuitively, making the horizontal size of **evolving** Northern Hemisphere Pleistocene ice sheets to be a constant, is strange, but from the basics of ice physics it is simply incorrect: H and L are not independent, and H is proportional to L^(1/2) (e.g., Verbitsky, 1992, Bahr et al, 2015). Taking this into account may dramatically change the dynamical properties of your governing equation (1).*

We understand your concern here and would completely agree except that $L$, in equation (1), does not represent the horizontal size of the ice sheet. We recognize that our current designation of L in that equation is incorrect and has understandably misled you. Indeed, in order to estimate the divergence of the flux between two horizontal points of an ice sheet, the local slope should be estimated by the inherent relationship between $H$ and $L$ (a given profile). It would therefore be incorrect to use a constant $L$. However, this is not what we are doing here. Under our (spatially) adimensional approach what Equation (1) says is simply: "The evolution of ice thickness (intended to be characterizing the mean state of the whole ice sheet) is the surface mass balance minus the ice discharge into the ocean". The latter would be captured in a 3D model by the ice flux at the grounding line, which is herein written as $v \cdot H/L$, where $L$ should be understood as a scale adjustable parameter and represents the length of the boundary between the ice sheet and the ocean, hereafter $L_{\mathrm{ocn}}$. To avoid misunderstanding, we will replace L in this equation by $L_{ocn}$ and the discharge will be

$$q = v \cdot \frac{H}{L_{ocn}} \quad \text{(Eq. CC1.1)}$$

However, we realize that when we introduce the dimensionalization of the horizontal derivatives, employing just $H/L$ with a fixed L can be problematic since the driving stress becomes quadratic with the ice thickness (equation 10). This is clearly not the case in real ice sheets (e.g. figure 4b from

Morlighem et al., 2013). Thus, we will include in the text a variable $L$ in Equation 10, following your recommendation of using

$$L = c \cdot H^2. \text{ (Eq. CC1.2)}$$

We have now implemented these changes in our model. Below follows a brief diagnostic and prognostic analysis of these changes. We use a range of coefficients $c$ in Eq. (CC1.2) from 0.2 (based on Vatnajökull ice cap) to 0.9 (based on Antarctic ice sheet). As can be shown in Figure CC1.1 below, in the case with fixed $L$, the driving stress leads to very high values, only exceeded for the extreme value c = 0.2 in about half of each cycle.

[Figure]

**Figure CC1.1.** Diagnosed $L$, $\tau_d$ and $q$ for a fixed $H$ evolution. Colors indicate different values of the coefficient in $L = c \cdot H^2$. Note that the black line corresponds to the case in which we kept $L$ constant.

We next show the effect of these changes on the AGING configuration presented in our manuscript for c = 0.9. As shown in Figure CC1.2, we can converge to the previous results, now with a dynamic L. The proposed changes in ice dynamics only slightly displace the parameter space. Therefore, we can say that despite the fact that we will need to recalibrate the simulations shown in the paper, our conclusions remain unchanged. We thank you for giving us the opportunity to correct it, and kindly invite you to proceed to the reading of the results and following sections.

[Figure]

**Figure CC1.2.** Simulations using AGING configuration for $c = 0.9$ (red) and our previous results with fixed $L$ (black).

*Few comments which are less significant, but may be you consider them helpful:*

1. *Your model is not "adimensional" as you claim in your PACCO abbreviation. For example, your equations have dimensional H, measured in meters, and dimensional time, measured in seconds. You didn't attempt to make your system adimensional. You simply do not resolve space.*

You are right in saying that PACCO is not an "adimensional" model but rather a "spatially adimensional" or a "zero-dimensional" model. We will clarify this in the revised version of the manuscript.

2. *It would be very helpful, if you present your final equations all together in one place instead of forcing a reader like me to do this work and substitute let say tau_d into v_d into v into q into dH/dt. At this point, the model description looks like a verbalized computer code (that I suspect it is).*

The gradual increase in complexity that structures the paper is fully intentional and aims to show the influence of each process on the periodicity and shape of the glacial cycles. We believe that gathering the equations as you suggest is in fact much closer to a "verbalized computer code" than our approach, which attempts to guide the reader through the model in a practical way. Additionally, we have to note that describing all the terms that participate in the ice dynamics (for example) in separated equations is the common practice. Otherwise, the following examples of the literature (e.g. Cuffey and Paterson (2010); Huybrechts and Oerlemans (1988); Quiquet et al. (2018); Pattyn (2017); Robinson et al. (2020)) should also be considered as "verbalized computer codes" (that we suspect are not).

3. *Since you position your model as a physical model, it would be nice if you explain the basic physical nature of the governing equations. For example, I was able to figure out that your equation (1) is scaled kinematic condition on the ice sheet free upper surface, but I am not sure I can explain with the same confidence equation (29) especially when Q is not defined.*

We will make sure to describe this properly in the revised versions of the manuscript. On the other hand, we will address your question about Equation (29) in CC2 below.

4. *Line 35: "However, most of them rely on very mathematical approaches and include artificial or imposed thresholds and trends (Paillard, 1998; Paillard and Parrenin, 2004; Gildor and Tziperman, 2001; Verbitsky et al., 2018; Ganopolski, 2024)." First, what "very mathematical" means? And, second, I am not aware about "artificial or imposed thresholds and trends" in Verbitsky et al (2018).*

In the revised version of the manuscript, we will refer more explicitly to the articles assuming the existence of thresholds and trends, which is what we understand to be a mathematical approach. In this context, we understand that "very mathematical" is not the right term and will resolve this in the revised version of the manuscript.

We emphasize that most of the processes resolved in those models are specific to the proposed hypothesis to explain glacial-interglacial variability. In contrast, PACCO solves a whole set of physical processes, covering various hypotheses and allowing to compare competing processes.

5. *And finally, a bit of funny thing. Your introduction begins with the sentence that makes a reader to believe that Paillard in 2001 and Ganopolski in 2024 introduced glacial-interglacial variability to the world. With all due respect to celebrated scientists, I would suggest someone like Agassiz to be mentioned there. It would also help to explain (next sentence) how Milankovitch was able to offer his theory before them.*

We kindly thank you for your suggestion. When we wrote the manuscript we did not intend to do a complete summary or review of the history of glacial cycles, but in fact it would be a nice addition to the introduction.

*Respectfully,*

*Mikhail Verbitsky*

*References*

*Bahr, D. B., Pfeffer, W. T., and Kaser, G.: A review of volume-area scaling of glaciers, Rev. Geophys., 53, 95–140, doi:10.1002/2014RG000470, 2015.*

*Birchfield, G.E. and Weertman, J., 1978. A note on the spectral response of a model continental ice sheet. Journal of Geophysical Research: Oceans, 83(C8), pp.4123-4125.*

*Chalikov, D.V. and Verbitsky, M.Y., 1984. A new Earth climate model. Nature, 308(5960), pp.609-612.*

*Crucifix, Michel. "Oscillators and relaxation phenomena in Pleistocene climate theory." Philosophical Transactions of the Royal Society A: Mathematical, Physical and Engineering Sciences 370, no. 1962 (2012): 1140-1165.*

*Verbitsky, M.Y. Equilibrium ice sheet scaling in climate modeling. Climate Dynamics 7, 105–110 (1992). https://doi.org/10.1007/BF00209611*

**CC2**

*I think I owe you a more explicit explanation of my discomfort with Equation (29).*

*It is a common knowledge that for typical ice sheets the Peclet number (Pe) is of order of 10. This means that temperature advection dominates heat diffusion and an ice-flow trajectory has a near-constant temperature determined by its value on the upper free surface of the ice sheet (e.g., Grigorian et al, 1976, Morland, 1984, etc). The thickness of the basal boundary layer where, instead, the heat diffusion begins to dominate, is proportional to Pe^(-1/2)\*H and is about 100 m. The timescale of the upper-surface temperature "delivery" to the basal boundary layer is the same as the timescale of ice growth. Your equation (29) seems to describe the heat balance of such basal boundary layer. Its thickness H_b = 10 m implies that you assume Pe to be even larger than 10. Nevertheless, the mechanism of delayed cold ice delivery to the bottom layer is absent in equation (29) and replacing it with conduction term (34) is very difficult to justify. Obviously, the absence of the vertical-temperature-advection timescale may have significant implications for the entire system dynamical properties.*

*Respectfully,*

*Mikhail Verbitsky*

*References*

*Grigoryan, S. S., M. S. Krass, and P. A. Shumskiy. "Mathematical model of a three-dimensional non-isothermal glacier." Journal of Glaciology 17, no. 77 (1976): 401-418.*

*Morland, L. W. "Thermomechanical balances of ice sheet flows." Geophysical & Astrophysical Fluid Dynamics 29, no. 1-4 (1984): 237-266.*

Thank you very much for clarifying your concern about Equation (29). Our approach was to keep the thermodynamics as simple as possible in order to characterize how different temperate base timescales influence the basal dynamics in the ice sheet. Thus, we assumed a heat balance between heat fluxes into the base ( $h_{geo}$ and $h_{drag}$ ) and out of it. Therefore, we assumed cooling given by the temperature

difference between the atmosphere and the base ($h_{\mathrm{cond}}$). Note that here, we understand "base" as a layer of thickness $H_b$. We were perhaps a bit careless when assigning to it a fixed value of 10 m without checking the Peclet number relation $H_b = Pe^{-1/2} \cdot H$, which would indeed yield a much larger value. We thus acknowledge your comment. Lastly, we had eliminated the advection term because its contribution was negligible and because we understood that its effect was accounted for by $h_{\mathrm{cond}}$, in the sense that the air temperature needs more time "to be delivered" to the base if the ice sheet is thicker. However, we understand that your more rigorous approach needs to be taken into account. Therefore, in the revised version of the paper we will explicitly include an advective term, $h_{\mathrm{adv}}$, and change Eq. (30) to

$$\frac{dT_{\mathrm{ice}}}{dt} = \frac{Pe^{1/2}}{c_{\mathrm{ice}} \cdot \rho_{\mathrm{ice}} \cdot H} \cdot (h_{\mathrm{geo}} + h_{\mathrm{drag}} + h_{\mathrm{cond}} + h_{\mathrm{adv}})$$

where

$$h_{\mathrm{cond}} = k_T \cdot \frac{T - T_{\mathrm{ice}}}{H \cdot (1 - Pe^{-1/2})},$$

$$h_{\mathrm{adv}} = c_{\mathrm{ice}} \cdot \rho_{\mathrm{ice}} \cdot \dot{s} \cdot \frac{T - T_{\mathrm{ice}}}{Pe^{1/2} - 1},$$

and by definition,

$$Pe = \frac{H - H_b}{k_T} \cdot \rho_{\mathrm{ice}} \cdot c_{\mathrm{ice}} \cdot \dot{s} \simeq \frac{H}{k_T} \cdot \rho_{\mathrm{ice}} \cdot c_{\mathrm{ice}} \cdot \dot{s}$$

Note that we removed the dependence on $H_b$ since it entails a second order variation on the heat fluxes (given that the dependency of $H_b$ on the Peclét number is already contained, e.g. $Pe^{-1/2}$ in the denominator) and we let the Peclet number evolve according to $H$ and the vertical velocity scale, that is assumed to be the snowfall $\dot{s}$.

We show in Figure CC2.1 below the THERM configuration applying the new equations for the same set of parameters. We have taken the advantage of improving the thermodynamics, following your suggestion, so we can now use a value of $h_{\mathrm{geo}}$ of 50 mW/m² (instead of the former 5 mW/m²), much closer to observations for North America and Europe. The 100 kyr periodicity is produced but still lacks appropriate time and shape. Thus, our conclusions regarding thermodynamics do not substantially change. The reason for this is that, under the THERM experiments, what ultimately controls the timing and amplitude of the deglaciation, is the required time for the base to become temperate, allowing the enhancement of basal sliding. With our former thermodynamics we were already exploring the phase space of this phenomenon (mainly through different values of $tau_{\mathrm{kin}}$). Yet we very much appreciate your comments and suggestions which we believe are contributing to improve our model.

[Figure]

**Figure CC2.1.** Comparative plot between the old (blue) and the new (orange) THERM configuration.

We now summarize the main changes that we consider to be implemented in the document:

1. We will add a variable for the typical horizontal spatial scale of the ice sheet following your suggestion $L = c \cdot H^2$ for the driving stress (Equation 10). However, we will keep $L_{\mathrm{ocn}}$ as a constant when calculating ice discharge (Equation 3).
2. We will clarify some terms and add some references that we consider relevant thanks to your comments.
3. We will modify the thermal balance in the basal boundary layer equations.

Finally, we would like to emphasize again that even though these comments are useful and have changed a few things in the manuscript (requiring recalibration in some cases), our results are qualitatively very similar and thus the conclusions are robust.

We hope our answers clarify your concerns.

Best regards,

Sergio Pérez-Montero et al.

References:
Cuffey, K. M., & Paterson, W. S. B. (2010). The physics of glaciers. Academic Press.

Huybrechts, P., & Oerlemans, J. (1988). Evolution of the East Antarctic ice sheet: a numerical study of thermo-mechanical response patterns with changing climate. Annals of glaciology, 11, 52-59.

Morlighem, M., Seroussi, H., Larour, E., & Rignot, E. (2013). Inversion of basal friction in Antarctica using exact and incomplete adjoints of a higher‑order model. *Journal of Geophysical Research: Earth Surface*, *118*(3), 1746-1753.

Quiquet, A., Dumas, C., Ritz, C., Peyaud, V., & Roche, D. M. (2018). The GRISLI ice sheet model (version 2.0): calibration and validation for multi-millennial changes of the Antarctic ice sheet. Geoscientific Model Development, 11(12), 5003-5025.

Pattyn, F. (2017). Sea-level response to melting of Antarctic ice shelves on multi-centennial timescales with the fast Elementary Thermomechanical Ice Sheet model (f. ETISh v1. 0). The Cryosphere, 11(4), 1851-1878.

Robel, A. A., DeGiuli, E., Schoof, C., & Tziperman, E. (2013). Dynamics of ice stream temporal variability: Modes, scales, and hysteresis. *Journal of Geophysical Research: Earth Surface*, *118*(2), 925-936.

Robinson, A., Alvarez-Solas, J., Montoya, M., Goelzer, H., Greve, R., & Ritz, C. (2020). Description and validation of the ice-sheet model Yelmo (version 1.0). Geoscientific Model Development, 13(6), 2805-2823.

Verbitsky, M. Y., Crucifix, M., & Volobuev, D. M. (2018). A theory of Pleistocene glacial rhythmicity. *Earth System Dynamics*, *9*(3), 1025-1043.

**Response to Mikhail Verbitsky comment (CC3)**

Dear Mikhail Verbitsky,

Thanks again for your comments. We hereafter respond point by point:

*Dear Sergio Pérez-Montero, Dear Colleagues,*

*Thank you for considering my comments.*

*1. Unfortunately, your equation CC1.1 is also problematic because it implies that ice sheets that do not*

*have a boundary with an ocean ($L_{\mathrm{ocn}} = 0$) have ice discharge $q \to \infty$.*

We disagree. We are not trying to simulate any ice sheet with equation CC1.1. We are aiming at estimating, in a spatially adimensional manner, the ice discharge of past Northern Hemisphere ice sheets, whose potential contact with the ocean is determined by their geographical distribution. Perhaps, $L_{\mathrm{ocn}}$ should rather be called "potential" or "maximum ocean boundary". And, the way of interpreting $L_{\mathrm{ocn}}$ is the following: according to equation CC1.1 and a given positive surface mass balance: an ice sheet with a low $L_{\mathrm{ocn}}$ would need a high ice discharge in order to be in equilibrium ( $dH/dt = 0$). Conversely, an ice sheet with a greater contact with the ocean (a high $L_{\mathrm{ocn}}$), would require (and show) a smaller ice discharge to be in equilibrium for that same surface mass balance. In any case, the relevant consideration regarding equation CC1.1 is whether the ice discharge $q$, scales well with $v \cdot H$ or some other relationship is better. We will show that indeed $q = k_1 \cdot v \cdot H$ is a good approximation, $k_1$ simply being $1/L_{\mathrm{ocn}}$. Therefore fixing $L_{\mathrm{ocn}}$ to a non-zero value is entirely justified (see below).

*Let us return to the basics. The total mass balance of an ice sheet can be written as*

$$\frac{\partial(HS)}{\partial t} = \dot{m}S - \oint_{\gamma} \int_{0}^{h_g} v_g dz dy$$

*Here $S$ is the area of an ice sheet, $\gamma$ is the grounding line, $h_g$, $v_g$ are ice thickness and normal velocity*

*at the grounding line, correspondingly. Indeed, the parameterization of the term $\oint_{\gamma} \int_{0}^{h} {}_g v_g dz dy$ is a*

*difficult task because at the grounding line ice thickness $h \to 0$, but ice slope $\partial h/dx \to \infty$.*

The ice thickness at the grounding line does not tend to 0. By definition the ice thickness must not be 0 there, otherwise it would not be a grounding line. Furthermore, the ice thickness at the grounding line (the mean value in Antarctica is about 220 m) is generally very similar to its surroundings, particularly in ice streams, because of the quite flat surface, and can sometimes be even higher than at its upstream vicinity. Also, the ice slope at the grounding line tends to be infinite only for the cases in which there is not downstream floating ice at all, which is a very rare case in Antarctica and unusual in Greenland. We believe this misinterpretation ($h \to 0$ and the slope to infinite at the grounding

line) is inherited from a rigid view of how the spatial profile of an ice sheet must be, and it frames your concerns regarding our chosen approach for the ice dynamics.

*You are not the first ones who face this challenge. For example, in 1980, as a PhD student, I had to present my work*

*to G. I. Barenblatt, and I was honored to be schooled by him about this term. I am not going to advocate*

*here necessarily for the parameterization that was adopted as the result of this conversation (we can*

*always discuss it off-line if needed), but if you want to continue to scale the above mass balance the same*

*way as you did before, then your mass discharge q should be proportional to*

$$q \sim \frac{L_{\mathrm{ocn}}}{L^2} v H \qquad \qquad \text{CC3.1}$$

Your proposed equation has, in turn, a really problematic feature: that the ice discharge will dramatically increase when $H$ is small. Because $L = c \cdot H^2$, your suggested equation for $q$ will go as $v/H^3$, which we believe is not appropriate. Nevertheless, we have tested both the validity of our former equation and the proposed one for $q$ by running a Northern Hemisphere glacial cycle with a state-of-the-art 3D thermomechanical ice sheet model, Yelmo (Robinson et al., 2020). The cycle is driven by a glacial index (e.g. Zweck and Huybrechts, 2005; Banderas et al., 2018) that weights the temperature and precipitation anomalies given by the PMIP3 climate models. Our simulation is shown in Figure CC3.1 in terms of the relative sea level elevation (RSLE) from 120 kyr BP to present day and four snapshots of ice velocities and ice thickness. This simulation was intended to be representative of the last glacial cycle (but not particularly tuned) and uses the same physics that Moreno-Parada et al. (2023), which shows good results in reproducing the last glacial maximum.

[Figure]

**Figure CC3.1.** Last glacial cycle simulated using Yelmo. Here we represent the relative to present day sea level elevation (RSLE) and different snapshots marked with their respective symbols of ice velocity (coloured) and ice elevation (the lines represent elevation isolines every 1000 m).

Using the ice velocities and thickness simulated by Yelmo we can diagnose the ice discharge using the two methods explained above (the current PACCO approach and your suggested equation (Eq. CC3.1)), and compare them to the simulated ice discharge by Yelmo, $q_{\text{Yelmo}}$, (Figure CC3.2). In the calculation we set the constants $k_1$ and $k_2$ so the diagnosed $q$ are in the best possible agreement with the simulated ice discharge $q_{\text{Yelmo}}$. As can be seen in Figure CC3.2, our approach shows a diagnosed $q$, which is much closer to $q_{\text{Yelmo}}$. Using the suggested Eq. CC3.1 implies an underestimation of the ice discharge for half of the cycle. Note that, even setting a minimum value of $H$ to 700 m (to avoid an explosion of $q$ when $H$ is small), your suggested equation clearly overestimates q at the beginning and the end of the cycle and underestimates it from the inception to the deglaciation. In conclusion, our approach is appropriate to simulate the mean state of the ice discharge of the ice sheet.

[Figure]

**Figure CC3.2.** (black) ice discharge calculated in Yelmo compared with the diagnosed ice discharge using (blue) PACCO formulation and (orange) the proposed formulation based on the ice sheet profile.

*"2. I am afraid my advocacy of the vertical temperature advection has been misunderstood. Within the basal boundary layer the vertical temperature advection is indeed negligible. What I meant is that regional air temperature $T$, that goes into calculations of your $h_{\text{cond}}$ is not instant $T(t)$ but it is temperature "from the past", i.e., $T = T(t - \tau)$ where $\tau = H/\dot{s}$ is the vertical-temperature-advection timescale, i.e. time of the temperature "delivery" from the ice-sheet free surface to the upper boundary of the basal boundary layer.*

*Respectfully,*

*Mikhail Verbitsky"*

Thanks for your clarification. We think that the delay is somewhat captured by both the Péclet number and $\tau_{\text{kin}}$. On one hand, we now let the advective regime determine the basal boundary layer thickness following your suggestion $H_b = H \cdot Pe^{-1/2}$. On the other hand, our sensitivity analysis (shown in the current version of the manuscript) shows that we explore the enhancement of basal sliding across a large range of kinematic wave times, $\tau_{\text{kin}}$. And we find that these timescales represent the delay you propose. The explored phase space would not change substantially given that $\tau_{\text{kin}}$ already captures the relaxation effect in the "temperature delivery". Ultimately, playing with thermodynamics aims to

capture the phenomenon by which the base becomes temperate, and explore the impact of it being higher, lower, faster or slower, in terms of its effects on ice sliding activation. In short, we aim at showing how a thermodynamically activated basal sliding affects deglaciations. And that is already largely explored in the current version of the manuscript and will be even more robust thanks to your comments.

This process has helped us to refine the manuscript and gain further confidence in our formulations, which are of course, just one of many possibilities. We believe PACCO represents a useful model for exploring glacial cycles and hopefully this is clear through its presentation.

Respectfully,

Sergio Pérez-Montero et al.

References

Banderas, R., Alvarez-Solas, J., Robinson, A., & Montoya, M. (2018). A new approach for simulating the paleo-evolution of the Northern Hemisphere ice sheets. *Geoscientific model development*, *11*(6), 2299-2314.

Moreno-Parada, D., Alvarez-Solas, J., Blasco, J., Montoya, M., & Robinson, A. (2023). Simulating the Laurentide ice sheet of the Last Glacial Maximum. *The Cryosphere*, *17*(5), 2139-2156.

Robinson, A., Alvarez-Solas, J., Montoya, M., Goelzer, H., Greve, R., & Ritz, C. (2020). Description and validation of the ice-sheet model Yelmo (version 1.0). Geoscientific Model Development, 13(6), 2805-2823.

Zweck, C., & Huybrechts, P. (2005). Modeling of the northern hemisphere ice sheets during the last glacial cycle and glaciological sensitivity. *Journal of Geophysical Research: Atmospheres*, *110*(D7).

---

## Referee Report (RR1)

Authors investigated Pleistocene glacial cycles that are known to be driven by the nonlinear response of the climate to solar forcing. Though complex models accurately simulate these cycles they are computationally expensive. Simpler conceptual models lack physical detail, though. The Physical Adimensional Climate Cryosphere mOdel (PACCO) used in this study aims to balance complexity and simplicity by focusing on the interaction between climate and Northern Hemisphere ice sheets. Authors show that PACCO effectively reproduces 100,000-year glacial cycles by incorporating ice-sheet dynamics, thermodynamics, and ice aging. The study reveals that ice aging and delayed isostatic response are key to matching geological records, making PACCOavaluable tool for studying glacial cycles.

Suggestions for the improvements:

Though I see the improvements made to this manuscript some of my comments are still not fully addressed and thus, I cannot recommend this article for a publication just yet. My comment about the energy spectra and peaks that we see at 60 and 100 kyears still stands; it is not a general observation that these peaks are present, but it really depends on the sliding parameter, and it mainly seems that the peaks are present for sliding parameter of $10^{-6}$ m yr$^{-1}$ Pa$^{-1}$, the question is why? Also, what happens above 100 kyr? The energy spectra are extremely dispersed, so what does that mean? Wouldn't we expect that at one point the energy goes to zero as we mainly see with the higher values of the sliding parameter? Additionally, I think that the authors put themselves in a trap by submitting Figure 1 in their rebuttal as I have an additional question now in connection to that figure. Here, it is not explained why there is such a big change between energy spectra of the ice thickness between different experiments. I think that this Figure 1 is now a great asset to the manuscript and it should be kept as it gives a great overview of the performed experiments and changes the complexity of the model brings to the results, still it needs to be made in a systematic way. Also, many of the figures need replotting as they are not done in a systematic way (looking at the axes, colours, etc.)

---

## Author Response (AR3)

**Response to: "Editor decision: Publish subject to minor revisions (review by editor)"**

Dear author, co-authors, first of all, sorry for the delay in my decision on your revised version of the ms submitted for publication in ESD on the conceptual ice sheet dynamics model PACCO and its application to evaluate the role of ice sheet dynamics and interactions with climate in explaining the peak signal in GIV at 100kyr. But now finally found the time to carefully check the revision and your response to the reviews. Overall it seems that you have handled well the last list of comments/feedback shared by the two reviewers but there is is still one remaining issue. Indeed once more again carefully reading the revision and then checking the figures in support of the statements on the model features, I also got confused and needed to go back and forth between those figures and the text. This has also to do with the fact that the initial use of the different colors referring to the different values of the sliding factor changed to other colors to assess the sensitivity to different parameters (e.g. Tau_kin). Here the interpretation of the figures would be more straightforward if this would be more explicitly mentioned in the figures captions (and possibly shortly in the introduction of the figure in the text). The main issue was with Figure 16 where it was not clear (not indicated in the figure captions) what the different colors reflected. A minor but essential feature to still fix also to fully appreciate the end result of the model development and application,

Laurens Ganzeveld

Dear editor,

Thank you for your comments. We have carefully reviewed the text again and added some sentences to explicitly say what each figure represents. All the changes can be found in the version comparison document but here we will summarize some of them:

> **line 190:** … over 800 kyr (Fig. 3 shows the simulations performed using different values of the sliding parameter Cs ), glacial inceptions …

> **Figure 13 caption:** … Time series obtained from the model using $\tau_{kin}$ (colors) …

> **line 360:** … A realization for different values of $\tau_\alpha$ is given in Fig. 15. …

> **Figure 15 caption:** Results of the AGING simulation. (a, c, e) Time series obtained from the model using different aging times $\tau\alpha$ . (b, d, f) Normalized periodograms obtained from the time series in the left column. We use a color bar for a range of periodicities in the GIV interval (10-120 kyr, from blue to green colors) and add some extreme values (1, 400, 1000, 10000 kyr), represented by lines with red shadings. Note that when normalizing spectra, series were cut off for periods larger than 200 kyr.

> **Figure 16:** We have moved the legend above the panels to clarify the meaning of the colors.

For the sake of being systematic, we have also placed the color bars and legends above the plots to facilitate the reading of the figures. Finally, some typos have been corrected and some missing doi have been added to the bibliography.

Sincerely,

Sergio Pérez-Montero et al.